# An entropy-based metric for assessing the purity of single cell populations

Baolin Liu [1,4], Chenwei Li[2,3,4], Ziyi Li[1], Dongfang Wang[1], Xianwen Ren[1] & Zemin Zhang [1,2,3 ✉]

Single-cell RNA sequencing (scRNA-seq) is a versatile tool for discovering and annotating cell types and states, but the determination and annotation of cell subtypes is often subjective and arbitrary. Often, it is not even clear whether a given cluster is uniform. Here we present an entropy-based statistic, ROGUE, to accurately quantify the purity of identified cell clusters. We demonstrate that our ROGUE metric is broadly applicable, and enables accurate, sensitive and robust assessment of cluster purity on a wide range of simulated and real datasets. Applying this metric to fibroblast, B cell and brain data, we identify additional subtypes and demonstrate the application of ROGUE-guided analyses to detect precise signals in specific subpopulations. ROGUE can be applied to all tested scRNA-seq datasets, and has important implications for evaluating the quality of putative clusters, discovering pure cell subtypes and constructing comprehensive, detailed and standardized single cell atlas.

[1] School of Life Sciences, BIOPIC and Beijing Advanced Innovation Centre for Genomics, Peking University, Beijing, China. [2] Peking-Tsinghua Centre for Life Sciences, Peking University, Beijing, China. [3] Analytical Biosciences Limited, Beijing, China. [4] These authors contributed equally: Baolin Liu, Chenwei Li. ✉email: zemin@pku.edu.cn

Tissues are complex milieus comprising various cell types and states with specialized roles[1]. Characterizing the property and function of each pure cell type is a long-standing challenge in biological and medical disciplines. The recent advances in scRNA-seq have transformative potential to discover and annotate cell types, providing insights into organ composition[2], tumor microenvironment[3], cell lineage[4], and fundamental cell properties[5]. However, the identification of cell clusters is often determined by manually checking specific signature genes, which are arbitrary and inherently imprecise. In

addition, different methods and even parameters used for normalization, feature selection, batch correction, and clustering can also confound the final identified clusters[6], thus motivating the need to accurately assess the purity or quality of identified clusters (Fig. 1a).

A pure cluster here is defined as a population where all cells have identical function and state without variable genes. The importance of purity assessment is particularly relevant for analyses that aim to discover novel pure subtypes and further detect the true biological signals. For example, signature genes specific to

**Fig. 1 The expression entropy model. a** Identifying pure cell subtypes in unsupervised single-cell data analysis. **b** The S–E plot of the Tabula Muris (droplet) dataset. Each point represents one gene. The relationship between S and E was fitted with LOESS regression for each gene. **c** The S–E plot of a T-cell dataset[27] obtained by Smart-seq2 protocol. **d** Accuracy in identifying differentially expressed genes on data simulated from both NB (left) and ZINB (right) distribution, with subpopulation containing 50% of the cells. The center line indicates the median AUC value. The lower and upper hinges represent the 25th and 75th percentiles, respectively, and whiskers denote 1.5 times the interquartile range. Discriminating power of genes selected by S–E model, HVG, Gini, M3Drop, SCTransform, Fano factor, and RaceID3 ("Methods") estimated by RF with 50 times cross-validation on both droplet-based dataset (**e**) and full-length-based dataset (**f**) listed in Supplementary Table 1. The classification accuracy was measured as the percentage of query cells that were assigned the correct label. The center line indicates the median classification accuracy. The lower and upper hinges represent the 25th and 75th percentiles, respectively, and whiskers denote 1.5 times the interquartile range. **g** Reproducibility of features of brain replicates (Supplementary Table 3). **h** ARI for the dataset comprising five cell lines[6] when different feature selection methods were used.

a pure subpopulation maybe mistakenly considered as the common signals of a mixture due to less guided clustering and annotation. The purity evaluation could therefore eliminate such misleading conclusions, potentially aiding our understanding of cellular function, state, and behavior. While pioneering approaches such as silhouette[7], DendroSplit[8], and distance ratio[9] have been devoted to determining the optimal number of identified clusters by calculating the ratio of within-cluster to intercluster dissimilarity, they are not comparable among datasets and have poor interpretability of cluster purity. For example, an average silhouette value of 0.7 indicates a fairly strong consistency for a given cluster, but it is still unknown whether this cluster is a pure population or a mixture of similar subpopulations especially when frequent dropout events occur.

The challenges presented by purity evaluation can be broadly addressed by investigating the number of infiltrating nonself cells and variable genes, which are suited to the intended areas of unsupervised variable gene detection. Given its importance, diverse methods[10] have been proposed for the quantification and selection of highly variable genes. In particular, scran[11] aims to identify variable genes by comparing variance to a local regression trend. However, the over-dispersion, coupled with the high frequency of dropout events, would often result in the selection of many lowly expressed genes[12]. Alternatively, although Gini coefficient[13] could be used to quantify the variation in gene expression, it is specially designed for rare cell-type identification. New probabilistic approaches for variable gene selection using dropout rates have also been recently adapted[12], with the advantage of supporting both pseudotime analysis and discrete clustering, but their usage of dropout metric hinders the capturing of the global distribution of gene expression. Although highly informative genes can also be determined by inspecting their weights during multiple iterations of dimensionality reduction[14], such ad hoc approaches are computationally intensive, requiring potentially orders of magnitude more time than methods like HVG and M3Drop.

Here, we present an entropy-based model to measure the randomness of gene expression in single cells, and demonstrate that this model is scalable across different datasets, capable of identifying variable genes with high sensitivity and precision. Based on this model, we propose the Ratio of Global Unshifted Entropy (ROGUE) statistic to quantify the purity or homogeneity of a given single-cell population while accounting for other technical factors. We demonstrate that the ROGUE metric enables accurate and unbiased assessment of cluster purity, and thus provides an effective measure to evaluate the quality of both published and newly generated cell clusters. Applying ROGUE to B cell, fibroblast, and brain data, we identified additional pure subtypes and demonstrate the application of ROGUE-guided analysis in detecting the precise biological signals. Our approach is broadly applicable for any scRNA-seq datasets, and is implemented in an open-source R package ROGUE (https://github.com/PaulingLiu/ROGUE), which is freely available.

## Results

**Overview of ROGUE.** As scRNA-seq data can be approximated by negative binomial (NB) or zero-inflated NB (ZINB) distribution[15,16], we considered the use of the statistic, S (expression entropy—differential entropy of expression distribution, as defined in "Methods"), to capture the degree of disorder or randomness of gene expression. Notably, we observed a strong relationship between S and the mean expression level (E) of genes, thus forming the basis for our expression entropy model (S–E model, Fig. 1b, c). Moreover, S is linearly related to E for the Tabula Muris dataset[2] as expected (Fig. 1b), which is

characteristic of current droplet experiments, hence demonstrating the NB nature of UMI-based datasets ("Methods"). For a heterogeneous cell population, certain genes would exhibit expression deviation in fractions of cells, leading to constrained randomness of its expression distribution and hence the reduction of S. Accordingly, informative genes can be obtained in an unsupervised fashion by selecting genes with maximal S-reduction (ds) against the null expectation ("Methods").

To provide a direct purity assessment of putative cell clusters or fluorescence-activated cell sorting (FACS)-sorted populations, here we take advantage of the wide applicability of S–E model to scRNA-seq data and introduce the quantitative measure, ROGUE ("Methods"). Intuitively, a cell population with no significant ds for all genes will receive a ROGUE value of 1, indicating it is a completely pure subtype or state. In contrast, a population with maximum summarization of significant ds will yield a purity score of ~0.

**S–E model accurately identifies informative genes.** To illustrate the performance of our model, we benchmarked S–E against other competing feature selection methods (HVG[11], Gini[13], M3Drop[12], SCTransform[17], Fano factor[18], and RaceID3[19]) on data simulated from both NB and ZINB distribution ("Methods"). For a fair comparison, we generated a total of 1600 evaluation datasets with subpopulations containing 50, 20, 10, or 1% of the cells, and used AUC as a standard to test the performance of each method. Notably, S–E model consistently achieved the highest average AUC and significantly outperformed other gene selection methods in all tested cases with varied subpopulation proportions or gene abundance levels (Fig. 1d and Supplementary Figs. 1 and 2). Although SCTransform is specially designed for UMI-based scRNA-seq data, it exhibited notable performance on ZINB-distributed datasets (Fig. 1d). As a tool to identify genes-specific to rare cell types, Gini showed increased performance when there were subpopulations accounting for <20% of the cells. In contrast, HVG performed better in the presence of cell subpopulations with a larger proportion (Supplementary Figs. 1 and 2).

To validate our unsupervised feature selection method in real datasets, we performed cross-validation experiments using random forest classifier (RF)[20]. We randomly sampled 70% cells from the original dataset as reference, and classified the remaining 30% cells, with clusters defined by the original authors ("Methods"). Intuitively, gene sets that enable higher classification accuracy are more biologically meaningful[21]. Using 14 previously published datasets derived from both droplet-based and full-length protocols (Supplementary Table 1), we demonstrated that our method consistently identified genes with greater ability of classification when different number (30–5000) of genes were selected (Fig. 1e, f and Supplementary Figs. 3 and 4). Specially, our S–E model showed notable superiority when fewer genes (30–100) were used, demonstrating its sensitivity. Taken together, these results suggest that genes identified by our model are more informative and biologically discriminating.

Since datasets derived from the same biological system are expected to have reproducible informative genes[12], we tested how our expression entropy model behaves using technical replicates from different tissues (Supplementary Table 2). Notably, genes identified by our S–E model were more reproducible when top 500–2000 genes were used (Fig. 1g and Supplementary Fig. 5a–c). In addition, we also considered four pancreatic datasets (Supplementary Table 3) derived from different technologies and labs. These real datasets are more complex than technical replicates as they included systemic nuisance factors such as batch effects. Despite substantial systematic differences, our model

consistently achieved high reproducibility scores (Supplementary Fig. 5d).

A major task of feature selection is to identify genes that are most relevant for biological heterogeneity, which can be applied to downstream clustering. We therefore evaluate the performance of S–E model in the context of unsupervised clustering with RaceID3[19], SC3[22], and Seurat[23]. Here we considered five publicly available scRNA-seq datasets with high-confidence cell labels[6,9,24,25] ("Methods"). These datasets include cells from different lines, FACS-purified populations, or well-characterized types (Supplementary Fig. 6 and "Methods"), and thus can be considered gold standards. To quantify the similarity between the clusters obtained by different clustering methods and the reference cell labels, we calculated the adjusted Rand index (ARI)[26], which is restricted to the interval [0, 1]. For the number of features, we considered the top 100, 500, 1000, or 2000 genes. Our results illustrated that S–E model provides the best performance in terms of ARI in these scenarios (Fig. 1h and Supplementary Fig. 7).

As some methods were optimized to detect rare cell types, we tested if the genes selected by our S–E model are effective in uncovering such rare subpopulations. To this end, we first simulated a scRNA-seq dataset ("Methods"), which contains three rare clusters (of 10, 30, and 20 cells, respectively) and two common clusters (of 1000 cells each), and clustered these cells with GiniClust2[18], RaceID3, as well as S–E model-based Seurat ("Methods"). Of note, all these three methods properly recapitulated the five cell clusters (Supplementary Fig. 8), indicating that S–E model-based Seurat is indeed effective for the recovery of both common and rare cell clusters. In addition to simulated data, we wondered how S–E model behaves in detecting real rare cell types. Since no gold standard is available for such cases, we considered four cell lines (A549, H2228, H838, and HCC827) of Tian et al.[6], and generated three common cell types (A549, H2228, and H838; of 500 cells for each) and one rare cell type (HCC827; of 20 or 10 cells, respectively) by down-sampling. Similar to the analysis of simulated data, all the three methods effectively identified both common and rare cell clusters when there were 20 cells of the rare cell type (Supplementary Fig. 9a–c). For the dataset with the rare cell type accounting for lower frequency (10 cells, 0.6% of total cells here), RaceID3 and GiniClust2 exhibited their superiority in uncovering the rare cell type as opposed to S–E model-based Seurat (Supplementary Fig. 9d–f). Thus, although S–E model is effective in uncovering rare subpopulations to a certain extent, methods specifically developed for this purpose, such as GiniClust2 and RaceID3, maybe more appropriate.

**Evaluation of robustness of ROGUE.** To test how sensitive ROGUE is to the choice of informative genes, here we considered two scRNA-seq datasets: a T-cell dataset sequenced with Smart-seq2[5] and a droplet-based dataset[2] (Tabula Muris). The results illustrated that the heterogeneity score (1-ROGUE) reached saturation when genes with significant $ds$ were selected (Supplementary Fig. 10), thus we used significant $ds$ to calculate ROGUE in the following analyses. We investigated the performance of ROGUE on 1860 cell populations simulated from both NB and ZINB distribution (2000 cells × 20,000 genes each), with 0.1–50% genes varied in a second cell type ("Methods"). A cell population with both fewer infiltrating nonself cells and varied genes would yield a high purity score, while a population with converse situation is expected to yield a low-purity score. It is evident that the ROGUE index decreased monotonically with the heterogeneity of cell populations (Fig. 2a, b and Supplementary Figs. 11 and 12). ROGUE performed well even when cell populations contained few varied genes (<1%) and infiltrating cells (<1%),

indicating ROGUE index provides a sensitive and unbiased measure in response to the degree of cell population purity. The usage of different values of the reference factor $K$ ("Methods") yielded vary similar results (Supplementary Fig. 13), suggesting that ROGUE is robust to the choice of parameter $K$ within a reasonable range.

To address the potential concern that the number of cells may represent an intrinsic challenge for S and ROGUE calculation, particularly if only a small number of cells are collected from given samples, we performed down sampling analysis to test how S was impacted by cell numbers. By calculating the Pearson correlations of S between the randomly down-sampled datasets and the entire datasets, we found the similarity values of >0.99 and demonstrated that our S and ROGUE calculation would not be affected by variation in cell number (Fig. 2c).

Sequencing depth can vary significantly across cells, with variation potentially spanning orders of magnitude[2], and hence contributes to a substantial technical confounder in scRNA-seq data. We sought to investigate whether ROUGE index can accurately assess the purity of single-cell population while accounting for this technical effect. As test cases, we simulated increasing molecular counts (sequencing depth) in a second mock replicate, with the fold change of gene expression means ranging from 2 to 100 (Fig. 2d and "Methods"). Despite the substantial technical effect, the mixture of each two simulated replicates is expected to be a pure cell population. Here we used silhouette to measure the degree of replicate-to-replicate differences. The results revealed ROGUE values of ~0.99–1 for each population consisting of two replicates, with silhouette values ranging from 0.25 to 0.75 (Fig. 2e, f and Supplementary Fig. 14a). Thus, ROGUE not only offers a robust and sensitive way to estimate the purity of single-cell population, but also accounts for the variation in sequencing depth.

**ROGUE accurately assesses the purity of cell populations.** To illustrate the applicability of ROGUE index to real data, we first considered an External RNA Controls Consortium (ERCC) dataset[24], which is a highly controlled experiment dedicated to understanding the technical variability. All 1015 droplets of this dataset received the same ratio of ERCC synthetic spike-in RNAs, hence no varied RNAs should be detected in principle. We referred to this dataset as an ideal case of pure cell population and found only one RNA with significant $ds$. Accordingly, this ERCC dataset achieved a ROGUE value of ~1 as expected, thus confirming its purity. Further, we investigated the fresh peripheral blood mononuclear cells (PBMCs) enriched from a single healthy donor[24]. The authors provided multiple cell types purified by FACS, and thus representing a suitable resource for purity assessment. These cell types in Fig. 2g, including CD4/CD8 naïve T cells and CD4 memory T cells, have been shown to be highly homogeneous populations[27], and were detected high ROGUE values (0.94–1) as expected. In contrast, both CD14 monocytes and CD34+ cells are mixtures of diverse subtypes[24] and received relatively low ROGUE values (~0.8; Fig. 2h), thus confirming their heterogeneity.

In addition to highly controlled datasets, it is also instructive to investigate how ROGUE index performs on pure subtypes identified by unsupervised clustering. Here we first considered six well-defined T-cell subtypes from human colorectal cancer[5], which were generated via the Smart-seq2 protocol. All these pure subtypes achieved high ROGUE values of >0.9 (Fig. 2i), versus 0.78 for complete data (Supplementary Fig. 14b). We next examined four dendritic cell (DC) subsets collected from human lung cancers[28] and sequenced with inDrop platform. Specially, tumor-infiltrating DC2 cells have been proven to be highly heterogeneous populations[29,30] and deviated substantially from the other

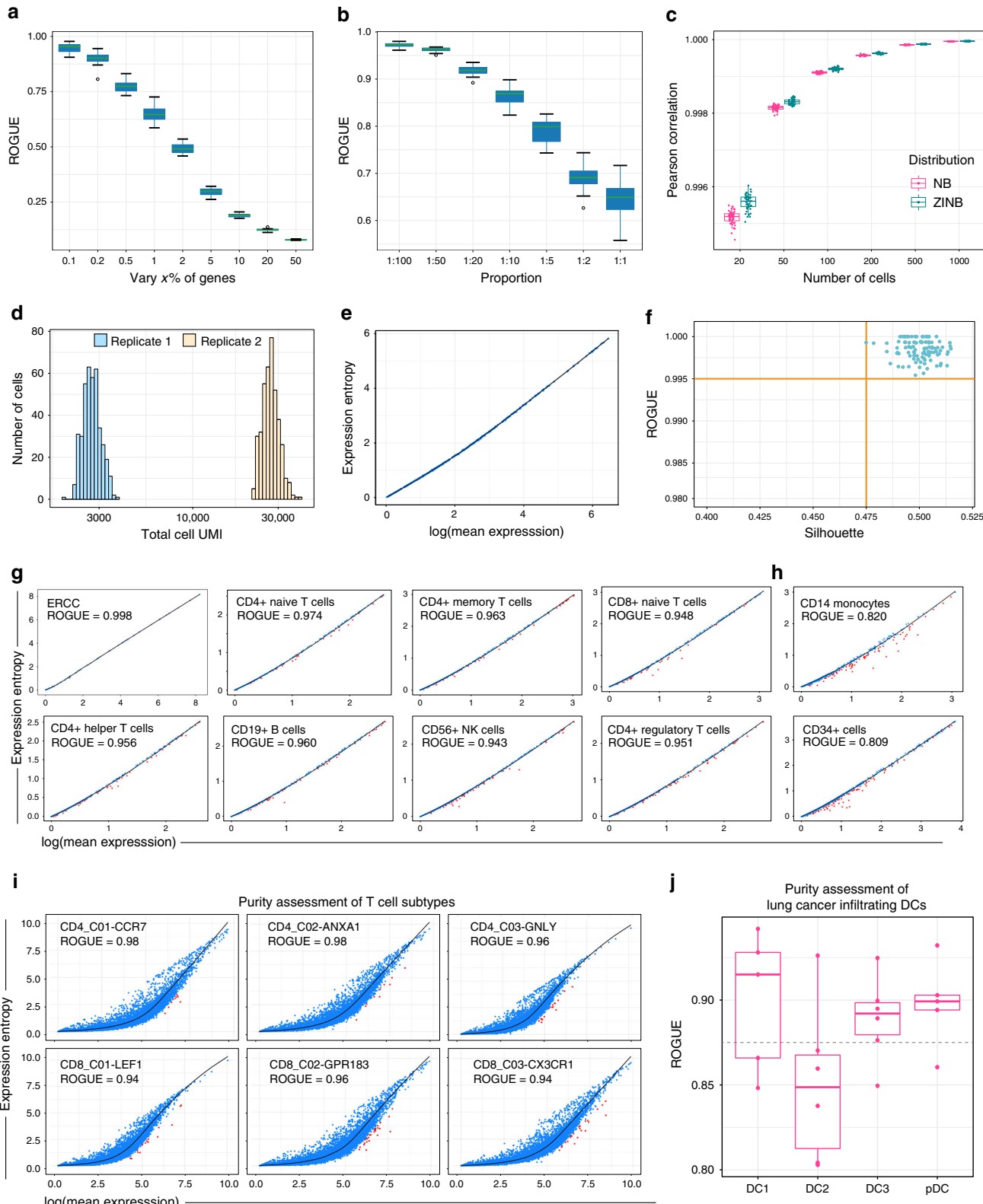

homogeneous cell types including DC1, LAMP3+ DC, and pDC (Fig. 2j). Taken together, these results illustrate that our ROGUE represents an effective and direct quantification of cell population purity without being affected by technical characteristics.

**ROGUE-guided analysis enhances cell-type identification.** We next evaluated the potential for ROGUE to guide clustering

analysis with silhouette, which investigates whether a certain clustering has maximized intercluster dissimilarity and minimized within-cluster dissimilarity. As a test case, we simulated a scRNA-seq dataset consisting of three cell types A, B, and C (see "Methods" for details), where cell types A and B were similar subtypes with 1% varied genes. We clustered this dataset into 2, 3, 4, and 5 subpopulations respectively by adjusting the resolution

**Fig. 2 ROGUE use and performance. a** The ROGUE index (reference factor $K = 45$) decreases monotonically with increasing varied genes in each simulated mixture consisting of two cell types (1:1). The center line indicates the median ROGUE value of $n = 50$ repeated simulations. The lower and upper hinges represent the 25th and 75th percentiles respectively, and whiskers denote 1.5 times the interquartile range. **b** The ROGUE values (reference factor $K = 45$) for the simulated mixtures with cell-type sizes ranging from 1:100 to 1:1. In each mixture, the number of varied genes was 1% of the total gene number ($n = 20,000$). The center line indicates the median ROUGE value of $n = 50$ repeated simulations. The lower and upper hinges represent the 25th and 75th percentiles respectively, and whiskers denote 1.5 times the interquartile range. **c** Pearson correlations of S between the randomly down-sampled datasets ($n = 50$ runs for each) and the entire datasets (2000 cells) simulated from both NB and ZINB distribution. The center line indicates the median correlation value. The lower and upper hinges represent the 25th and 75th percentiles respectively, and whiskers denote 1.5 times the interquartile range. **d** Sequencing depth distribution (total UMI counts across cells) for two simulated replicates. The replicate 2 has a sequencing depth ten times that of replicate 1. **e** The S–E plot of the mixture of replicates 1 and 2 is shown in **d**. **f** ROGUE values of $n = 100$ mixtures versus the silhouette values for every two replicates within individual mixtures. A high silhouette value indicates a substantial difference in sequencing depth between two replicates. **g, h** The S–E plots and corresponding ROGUE values of 10 cell populations from the PBMC dataset[24]. **i** Purity assessment of six human T-cell populations. **j** Purity evaluation of lung-cancer infiltrating DCs, with each point representing a patient. The center line indicates the median ROUGE value. The lower and upper hinges represent the 25th and 75th percentiles, respectively, and whiskers denote 1.5 times the interquartile range.

parameter in Seurat[23] (Fig. 3a), then evaluated the results by inspecting corresponding silhouette and average ROGUE values. Proper clustering of this dataset should result in three subpopulations, one for each cell type. However, silhouette received the maximum value when cell-type A co-clustered with B (Fig. 3b), i.e., when only two clusters were identified, suggesting that such measure is poorly interpretable for cluster purity as opposed to ROGUE, which reached saturation when there were three clusters (Fig. 3c). Repeating the simulation with varied differences in cell-type A, B, and C yielded equivalent performance for these two methods (Supplementary Fig. 15a–f). Such performance was also observed when different values of the reference factor $K$ were used (Supplementary Fig. 16). Since ROGUE can provide direct purity quantification of a single cluster and is independent of methods used for normalization, dimensionality reduction, and clustering, it could also be applied to guide the splitting (re-clustering) or merging of specific clusters in unsupervised clustering analyses.

To test how ROGUE could help the clustering of real datasets, we examined a previously reported dataset of cancer-associated fibroblasts (CAFs)[31] from lung tumors. CAFs have been reported to represent a highly heterogeneous population and may play a tumor-supportive role in the tumor microenvironment[32]. We found that the seven identified fibroblast clusters received low ROGUE values (Fig. 3d, e and Supplementary Fig. 17a). We therefore performed re-clustering analysis with the goal of exploring the extent of heterogeneity and identified a total of 11 clusters with a higher average ROGUE value (Fig. 3d, e). In addition to the two classical subtypes of CAFs (myofibroblastic CAFs and inflammatory CAFs), we also found the presence of antigen-presenting CAFs (apCAFs) that was characterized by the high expression of *CD74* and MHC class-II genes (Supplementary Fig. 17b). The apCAFs were firstly discovered as a fibroblast subtype in mouse pancreatic ductal adenocarcinoma (PDAC), but barely detectable in human PDAC without forming a separate cluster[33]. The considerable existence of apCAFs in lung cancer thus may indicate potential differences between different cancer types.

Furthermore, we noted that the myCAFs (AF_C02_COL4A1, ROGUE = 0.81) identified by original authors could be further segregated into three distinct subpopulations, including BF_C01_RGS5 (ROGUE = 0.84), BF_C02_ACTA2 (ROGUE = 0.87), and BF_C03_GPX3 (ROGUE = 0.94). Interestingly, the signature genes of AF_C02_COL4A1 described by original authors were actually specific to one of these three subpopulations, including *MEF2C* in BF_C01_RGS5 and *MYH11* in BF_C02_ACTA2 (Fig. 3f). Pathway analysis also revealed that the NOTCH signaling was activated in BF_C01_RGS5 (Fig. 3g) rather than a common signal of AF_C02_COL4A1[31]. Despite the considerable increase of

overall ROGUE index, BF_C00_AOL10A1, BF_C04_COL1A2, and BF_C05_PLA2G2A still received relatively low ROGUE values, thus deserving further investigation. Overall, ROGUE-guided analysis not only discovered novel cell subtypes, but also enabled the detection of the true signals in specific pure subpopulations.

**ROGUE-guided analysis identifies pure B cell subtypes.** B cells are key components in tumor microenvironment but have unclear functions in antitumor humoral response[34]. Here we investigated previously reported liver- and lung-tumor-infiltrating B cells[31,35] and found that they received relatively low ROGUE values (Fig. 4a). Thus, we applied further clustering analysis coupled with ROGUE to these B cells in an attempt to discover pure subtypes. A total of seven clusters were identified, each with its specific marker genes (Fig. 4b–d). Cells from the first B-cell subset, B_C0_JUNB, specifically expressed signature genes including *JUNB* and *FOS*, thus representing activated B cells[36]. The second subset, B_C1_TXNIP, showed high expression of glycolysis pathway genes (Supplementary Table 4), indicating its metabolic differences. *ACTB*, a gene involved in antigen presenting, was highly expressed in the third subset (B_C2_ACTB). Pathway activity analysis also revealed a strong antigen processing and presentation signal in this subset (Supplementary Table 4). The fourth cluster, B_C3_FCER2, characterized by high expression of *HVCN1* and genes involved in B-cell receptor signaling pathway (Supplementary Table 4), was largely composed of pre-activated B cells[37]. The fifth cluster, B_C4_MX1, predominantly composed of interferon-induced B cells[38], expressed high levels of *MX1*, *IFI6*, and *IFI44L*. The sixth cluster, B_C5_CD3D, expressed key markers of both T- and B-cell lineages (Fig. 4d), thus maybe the dual expressers (DEs)-like lymphocytes[39] or doublets. The remaining B cells, falling into the seventh cluster, B_C6_LRMP, exhibited high expression of *LRMP* and *RGS13*, indicative of the identity of germinal center B cells[40].

Both DEs/doublets-like and germinal center B cells exhibited low ROGUE values (Fig. 4e), but the limited cells did not permit further clustering. For germinal center B cells, we readily detected the high expression of proliferating marker genes, including *MKI67* and *STMN1* (Supplementary Fig. 18), in a fraction of these cells, thus explaining the heterogeneity to some extent. In contrast to these two clusters, we found ROGUE values of >0.92 for each of the remaining five clusters (Fig. 4e), demonstrating that they were all highly homogeneous B-cell subtypes. By calculating the ratio of observed to expected cell numbers with the chi-square test (RO/E), we noted that both B_C02_ACTB and B_C04_MX1 contained mainly cells from tumor, with RO/E values >1 (Fig. 4f). Similar analyses stratified by patient further confirmed this trend (Fig. 4g). Based on the independent TCGA

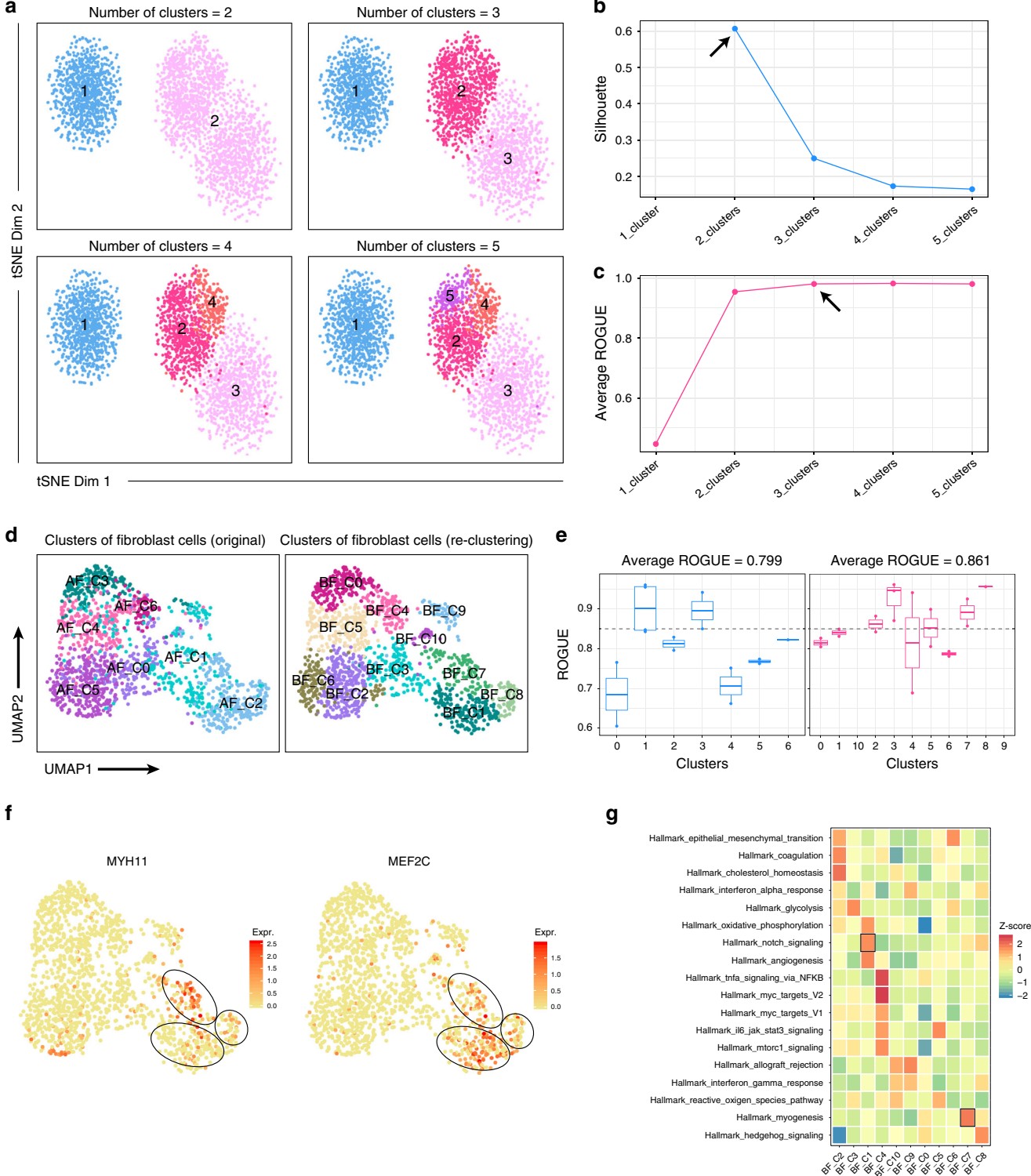

**Fig. 3 ROGUE enhances single-cell clustering and cell-type identification. a** t-SNE plots of a simulated dataset containing three cell types. Corresponding silhouette values (**b**) and average ROGUE values **(c)** when there were 2, 3, 4, and 5 putative clusters, respectively. **d** UMAP plots of lung-cancer-associated fibroblasts, color-coded by clusters in original paper (left; Supplementary Fig. 17a) and re-clustered labels (right). **e** ROGUE values of different clusters before (left) and after (right) re-clustering. Each point represents a patient. The center line indicates the median ROUGE value. The lower and upper hinges represent the 25th and 75th percentiles respectively, and whiskers denote 1.5 times the interquartile range. **f** UMAP plot of expression levels of *MYH11* and *MEF2C*. **g** Differences in hallmark pathway activities scored using GSVA.

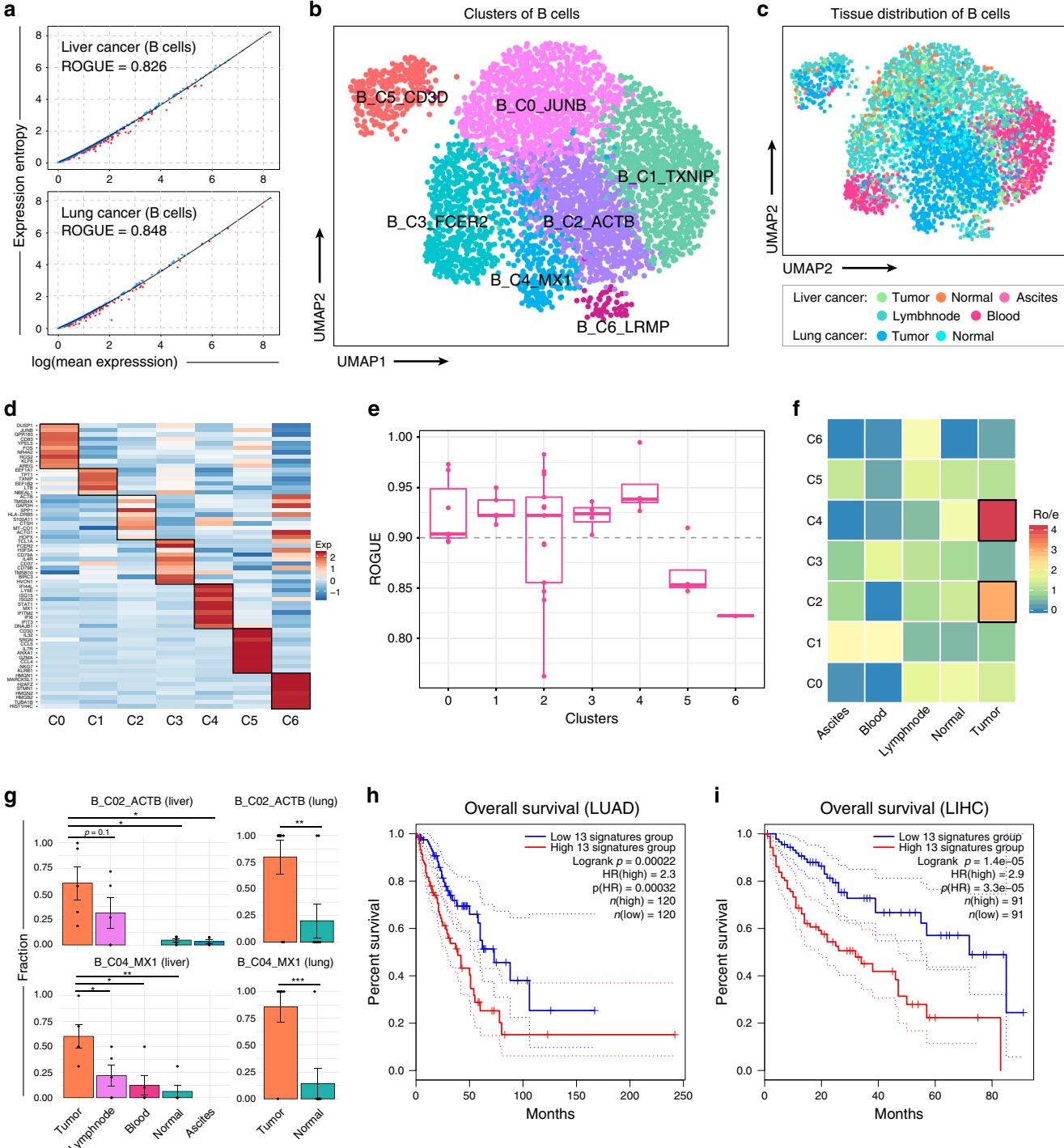

**Fig. 4 ROGUE-guided analysis in the identification of pure B-cell subtypes. a** The S–E plots and ROGUE values of liver- and lung-tumor-infiltrating B cells, respectively. UMAP plots of 4291 B cells, color-coded by their associated clusters (**b**) and tissues (**c**). **d** Gene expression heatmap of seven B-cell clusters. Rows denote marker genes and columns denote different clusters. **e** ROGUE values of seven identified B-cell subtypes. Each point represents a patient. The center line indicates the median ROUGE value. The lower and upper hinges represent the 25th and 75th percentiles, respectively, and whiskers denote 1.5 times the interquartile range. **f** Tissue preference of each B-cell subtype in liver cancer estimated by RO/E[27], the ratio of observed to expected cell numbers calculated by the chi-square test. **g** The average fractions of B_C02_ACTB and B_C04_MX1 in each patient across tissues, where error bars representing ±s.e.m. *p < 0.05, **p < 0.005, Student's t test. The Kaplan–Meier curves of TCGA LUAD (**h**) and LIHC (**i**) patients grouped by the 13 markers (Supplementary Table 5) of B_C02_ACTB.

lung adenocarcinoma (LUAD) cohort dataset, patients with higher expression of the marker genes of B_C02_ACTB (normalized by MS4A1; Supplementary Table 5) showed significantly worse overall survival (Fig. 4h). Such survival difference was also observed in TCGA liver hepatocellular carcinoma (LIHC) cohort

dataset (Fig. 4i). Thus, the clinical implication deserves further study to investigate what specific roles B_C02_ACTB cells play in tumor microenvironment. In summary, identifying pure subtypes with ROGUE-guided analysis could enable a deeper biological understanding of cell state and behavior.

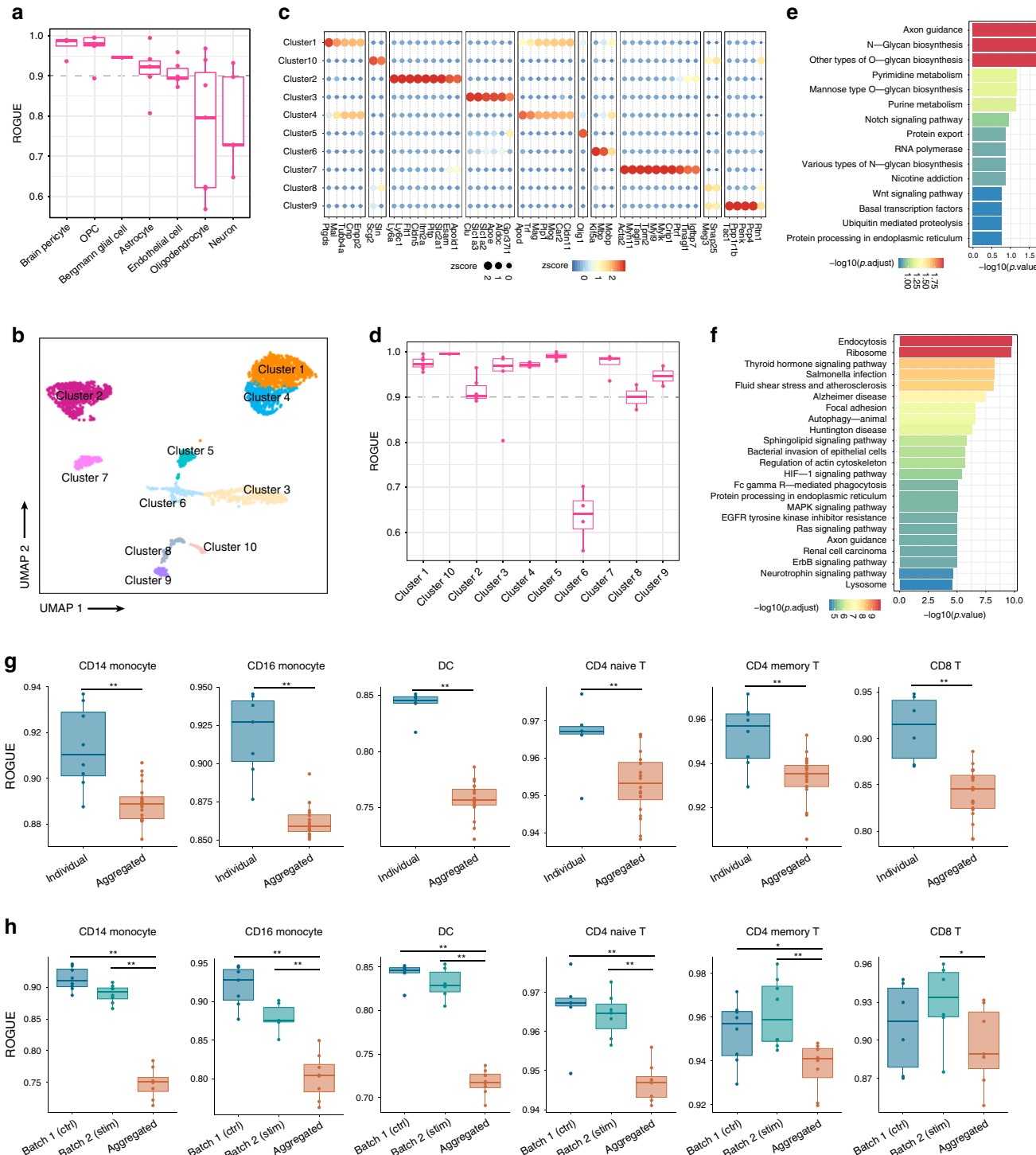

**Application to brain data and batch effect evaluation**. In addition to cancer data, we also demonstrated the application of ROGUE in analyzing the brain transcriptome dataset[2], which harbors a high degree of heterogeneity for those encapsulated cell classes. This dataset identified seven distinct cell types, of which oligodendrocyte and neuron cell types had low ROGUE values of <0.8, versus ~0.9–1 for the remaining five cell classes (Fig. 5a). We therefore applied further clustering guided by ROGUE to oligodendrocyte which is of enough cells ($n = 3401$), and identified ten refined cell subtypes, each with its specific marker genes (Fig. 5b, c). Except cluster 6, we found ROGUE values of ~0.9–1 for all the other nine clusters, suggesting their purity (Fig. 5d). To

investigate potential functions of these subtypes, we compared pathway activities and found considerable phenotypic diversity. For example, cluster 5 showed a strong signal of axon guidance signaling (Fig. 5e), while neurotrophin signaling pathway was highly activated in cluster 1 (Fig. 5f). This example further illustrates how ROGUE plays a key role in uncovering pure subpopulations.

To investigate if ROGUE is effective in evaluating the impact of batch effect, we studied a dataset of human PBMCs containing multiple distinct cell types[38]. Cells of this dataset were previously split into two groups—the interferon-beta (IFN-β)-stimulated group and the culture-matched control group, thus could be

**Fig. 5 The application of ROGUE in brain data and batch effect evaluation. a** ROGUE values of seven distinct brain cell types as defined by the original publication[2], with each point representing a sample. The center line indicates the median ROUGE value. The lower and upper hinges represent the 25th and 75th percentiles respectively, and whiskers denote 1.5 times the interquartile range. **b** UMAP plot of the ten identified clusters of oligodendrocytes ($n = 3401$), color-coded by their associated clusters. **c** Expression heatmap of cell-type-specific genes of the ten oligodendrocyte clusters. **d** ROGUE values of oligodendrocyte clusters. Each point represents a sample. The center line indicates the median ROUGE value. The lower and upper hinges represent the 25th and 75th percentiles, respectively, and whiskers denote 1.5 times the interquartile range. **e, f** Enriched pathways for cluster 5 (**e**) and cluster 1 (**f**), respectively. **g** ROGUE values were shown for batch 1 (the control group), batch 2 (the stimulation group), and aggregated cell population (batch 1 and batch 2) for each cell type. For fair comparison, we equalized the number of cells in each group by down-sampling. The center line indicates the median ROGUE value. The lower and upper hinges represent the 25th and 75th percentiles, respectively, and whiskers denote 1.5 times the interquartile range. *$p < 0.05$, **$p < 0.005$, Student's $t$ test. **h** ROGUE values for individual-specific cell populations and aggregated populations (all individuals). All cells used here were from the control group. Subsampling was performed to equalize the number of cells in each group. The center line indicates the median ROUGE value. The lower and upper hinges represent the 25th and 75th percentiles, respectively, and whiskers denote 1.5 times the interquartile range. *$p < 0.05$, **$p < 0.005$, Student's $t$ test.

considered as two batches. Then we applied ROGUE to assess the purity of each cell type (as defined by the original authors) in individual bathes as well as the aggregated cell population (batch 1 and batch 2), and found that ROGUE detected considerable purity reduction in the aggregated group (Fig. 5g).

As cells of this dataset were collected from eight unrelated individuals, we also tested how ROGUE behaves in estimating the variability (i.e., batch effect) among patients. Here we only used cells from the control group so that the evaluation would not be influenced by IFN-β perturbation. As expected, the aggregated cell populations of all individuals received significantly lower ROGUE values as opposed to patient-specific populations for each cell type (Fig. 5h). Thus, ROGUE offers a reasonable method for estimating the impact of batch effect.

## Discussion

Purity assessment of identified cell clusters is paramount to the interpretation of scRNA-seq data. This assessment is especially pertinent as increasingly rare and subtle cell subtypes are being uncovered. To address this computational challenge, we present the S–E model and demonstrate that this model is capable of identifying variable genes with high sensitivity and precision, and thus could be applied to both clustering and potentially pseudotime analyses. By taking advantage of the wide applicability of S–E model, we develop the statistic ROGUE to quantify the purity of single-cell populations. Through a wide range of tests, we demonstrate that our entropy-based measure, ROGUE, is broadly applicable for datasets from different platforms, protocols and operators, and able to successfully quantify the purity of singl-cell populations regardless of uncontrollable cell-to-cell variation.

When using ROGUE to assess the purity of four DC subtypes from human lung tumors, we found that DC2 was a heterogeneous population, which is consistent with previous findings[30]. Such heterogeneous populations like DC2 may have different properties and specialized roles in the cancer microenvironment, and could be assessed in a similar fashion with ROGUE. Accordingly, future studies could focus on these cell populations and hence may deepen our understanding of cellular origins of cancer. In addition, ROGUE addresses an important need in unsupervised single-cell data analyses, i.e., to effectively assess the quality of published or newly generated clusters. Often, unsupervised clustering may lead to under- or over-clustering of cells due to the lack of universal stands for clustering quality. By quantifying cluster purity with ROGUE before and after clustering or re-clustering, we were able to detect low-purity clusters and perform further analysis to discover pure subtypes. Improving the purity and credibility of the ever-increasing number of cell types is a mounting challenge with explosive efforts toward single-cell sequencing, and ROGUE could become a potential standard for judging the quality of cell clusters.

Our ROGUE-guided analysis on fibroblasts identified a novel subpopulation in lung cancer, apCAFs, which highly expressed *CD74* as well as MHC class-II genes and had a strong antigen-presenting signal. These cells have been speculated to deactivate CD4 T cells and decrease the CD8+ to Treg ratio in mouse PDAC[33], but have unclear role in the lung-cancer microenvironment, hence requiring further investigation. Moreover, when applying ROUGE to B-cell analysis, we found an interesting pure cluster B_C02_ACTB that displayed high expression of genes involved in antigen processing and presentation. Cells from this cluster were preferentially enriched in tumors and were associated with poor prognostic outcomes in both lung and liver cancer. We therefore hypothesize that these cells may contribute to immune suppression in the cancer microenvironment and hence curtail antitumor immunity, although further studies are required to define the roles of these cells. Such approaches for discovering novel or additional pure subtypes can also be extended to other published or newly generated scRNA-seq datasets.

When determining the purity of cell clusters, we recommend a ROGUE value of 0.9 as a suitable threshold, at which the number of infiltrating cells and varied genes is well constrained. But for low-quality data or continuous data, the threshold could be determined by considering the global ROGUE values. Although ROGUE can be very efficient and effective, we anticipate that additional extensions could enable enhanced performance, for example, assessing the purity of integrated cell populations from different protocols and platforms. Overall, our ROGUE metric provides a robust and direct measure for cluster purity in the presence of substantial technical confounders. We expect the ROGUE metric to be broadly applicable to any scRNA-seq datasets, and anticipate that our strategy will improve the rigor and quality of unsupervised single-cell data analysis.

## Methods

**Expression entropy model**. For droplet datasets, the observed UMI count can be modeled as a NB random variable, which also arises as a Poisson–Gamma mixture[41]

$$
\begin{aligned}
X_{ij} &\sim \text{Poisson}\,(s_j\lambda_{ij}) \\
\lambda_{ij} &\sim \text{Gamma}\,(\alpha_{ij}, \beta_{ij})
\end{aligned}, \tag{1}
$$

where $\lambda_{ij}$ represents the true expression value that underlies the observed UMI count $X_{ij}$ of gene $i$ in cell $j$, and $s_j$ denotes the size normalization factor in cell $j$. The $\alpha_{ij}$ and $\beta_{ij}$ are shape parameter and rate parameter respectively. Given the assumption that the shape parameter $\alpha$ is a constant across cells and genes, and that the rate parameter $\beta$ is a constant of gene $i$ across cells[41,42], $\alpha_{ij}$ and $\beta_{ij}$ can be expressed as $\alpha$ and $\beta_i$, respectively. Then the distributions can be recognized as: $\lambda_i \sim \text{Gamma}\,(\alpha, \beta_i)$ and $X_{ij} \sim \text{Poisson}\,(s_j\lambda_i)$. We denote

$$
X'_{ij} = \frac{X_{ij}}{s_j}, \tag{2}
$$

as the normalized expression of gene $i$ in cell $j$, and use $\mathbb{E}(X'_i)$ ($X'_i$ is the normalized expression assigned to gene $i$ and $\mathbb{E}(X'_i)$ is the expectation across cells) as the

moment estimation of $\lambda_i$. For the Gamma distribution, the rate parameter could therefore be calculated based on the maximum likelihood estimation

$$\beta_i = \frac{\alpha}{\lambda_i} = \frac{\alpha}{\mathbb{E}(X'_i)}. \tag{3}$$

To capture the degree of disorder or randomness of gene expression, here we considered the use of differential entropy defined as[43]

$$H(X) = -\int_{-\infty}^{+\infty} p(x) \cdot \ln p(x) dx, \tag{4}$$

where $X$ is a continuous random variable and $p(x)$ is the probability density function. Differential entropy is an extension of Shannon entropy, which is used to measure the average surprisal of a continuous probability distribution, and has shown notable performance in our supervised gene selection method E-test[44]. Specially, for the gamma distributed random variable $\lambda_i$, its differential entropy can be computed as

$$S_i = \alpha - \ln\beta_i + \ln\Gamma(\alpha) + (1-\alpha) \cdot \varphi(\alpha) = \ln\frac{\alpha}{\beta_i} + a = \ln\mathbb{E}(X'_i) + a, \tag{5}$$

where $\varphi$ is the digamma function, and $a = \alpha - \ln\alpha + \ln\Gamma(\alpha) + (1-\alpha) \cdot \varphi(\alpha)$ is a constant. Although other pioneering methods such as Scnorm[45], scran[46], and BASiCS[47] can be used to calculate size factors, we considered the library size normalization of each cell defined as the total UMI counts divided by the mean total UMI counts across cells[41]. Accordingly, the expectation of library size factor across cells is equal to 1. Given Eq. (2) and that the gene expression and library size are two independent random variables[42], for a given gene $i$, we have

$$\mathbb{E}(X_i) = \mathbb{E}(X'_i \times s) = \mathbb{E}(X'_i) \times \mathbb{E}(s) = \mathbb{E}(X'_i) \times 1 = \mathbb{E}(X'_i), \tag{6}$$

where $X_i$ is the observed expression assigned to gene $i$ and $s$ is the library size assigned to cells. Thus, for each cell type, the differential entropy of $\lambda_i$ could be computed as

$$S_i = \ln\mathbb{E}(X_i) + a. \tag{7}$$

We formulate the null hypothesis that there is only one Poisson–Gamma component for each gene in a given population ($H_0$) and thus the corresponding differential entropy can be calculated with Eq. (7). Then we assume that each cell represents its own cluster and use $X_{ij}$ as a moment estimation of the mean expression of such cluster. In this way, we define the entropy reduction of gene $i$ across $n$ cells as

$$ds_i = \text{differential entropy under } H_0 - \text{average actual differential entropy}$$
$$= \ln\mathbb{E}(X_i) - \frac{\sum_{j=1}^n (\ln X_{ij})}{n}, \tag{8}$$

which captures the degree of disorder or randomness of gene expression[44]. Given that genes under $H_0$ (non-variable genes) account for the major proportion, we fit the relationship between $\ln \mathbb{E}(X_i)$ and average actual differential entropy, and calculate corresponding residual as $ds_i$ to improve the performance (Fig. 1b, c). The significance of $ds$ is estimated based on a normal distribution approximation and is adjusted using Benjamini–Hochberg method. We also extended such procedure to full-length datasets and found that our approach consistently outperformed other gene selection methods (Fig. 1f and Supplementary Fig. 4).

**Data simulation**. We simulated droplet datasets with NB distribution. Mean gene abundance levels $E$ were sampled from the log-normal distribution

$$ln(E) \sim N(\mu, \sigma^2),$$

with parameters $\mu = 0$ and $\sigma = 2$. The number of transcripts for each gene were drawn from

$$N_{ij} \sim \text{NB}(E_i, r).$$

For each simulated dataset, the dispersion parameter $r$ ($r = \alpha$)[48] was set to a fixed value, ranging from 5 to 20 (Supplementary Fig. 1). In addition, we simulated full-transcript datasets with ZINB distribution. The dropout rates for each gene was modeled with the sigmoid function[49]

$$P_i \sim \text{sigm}(-(\gamma_0 + \gamma_1 E_i)),$$

with parameters $\gamma_0 = -1.5$ and $\gamma_1 = 1/\text{median}(E)$. Each simulated scRNA-seq dataset contained 20,000 genes and 2000 cells (Supplementary Fig. 2).

Differentially expressed genes were added in a fraction of cells (1–50%, Supplementary Fig. 1 and 2), with fold changes sampled from the log-normal distribution ($\mu = 0$ and $\sigma = 2$). Genes with a >1.5-fold decrease or increase in mean expression were considered as ground truth DE genes.

**Feature selection methods**. The HVG method[11] identifies variable genes by comparing the coefficient of variation squared to a local regression trend, and was implemented with the BrenneckeGetVariableGenes function in the M3Drop[12] package. In the Gini index model proposed in GiniClust[13], a gene is considered as informative if its Gini is higher than expected from the maximum observed expression. We copied the source code of original GiniClust

(GiniClust_Preprocess.R, GiniClust_Filtering.R and GiniClust_Fitting.R) (https://github.com/lanjiangboston/GiniClust/tree/master/Rfunction), and defined the Gini_fun function in our scripts to select genes. M3Drop uses dropout rates for variable gene selection and was implemented with the M3DropFeatureSelection function in the M3Drop package. The SCTransform method[17] selects genes with Pearson residuals from the regularized negative binomial regression and was implemented with the SCTransform function in Seurat package. In addition, we implemented the Fano factor method as used in the script GiniClust2_Fano_-clustering.R from GiniClust2[18]. The feature selection step in RaceID3[19], which selects genes with a second-order polynomial fit between the expression variance and log-transformed mean, was implemented according to the fitBackVar function in RaceID3.

**Datasets used for clustering-based evaluation**. To evaluate the performance of different feature selection methods in the context of unsupervised clustering, here we considered five publicly available scRNA-seq datasets. The first dataset[6] consists of five cell lines (A549, H1975, H2228, H838, and HCC827) and was sequenced with 10X Genomics protocol, with a total of 3918 cells. The second dataset was generated by the same study[6]. This dataset comprises three cell lines (H1975, H2228, and HCC827) and was sequenced with CEL-seq2 protocol. The third dataset[24] was created by processing multiple FACS-purified cell populations and was sequenced with 10X Genomics protocol. Considering that some populations such as CD8+ cytotoxic T cells were relatively heterogenous[24], here we only used CD19 B cells, CD4 naïve T cells, CD56 NK cells, and CD14 monocytes, which were readily distinguishable (Supplementary Fig. 6a). The fourth dataset contains cells from human pancreatic islet and was generated by Smart-seq protocol[25]. These pancreatic cell types including alpha, beta, delta, and gamma cells are well-characterized and have been shown to be distinct[23,44], thus were used for benchmarking (Supplementary Fig. 6b). The remaining dataset comprises multiple immune cell types[9], with cells sequenced by Smart-seq2 protocol. Although the cell labels in original publication were assigned using unsupervised clustering, cross-validation experiments revealed that the major cell types (macrophages, DCs, lymphocytes, and exhausted CD8 T cells) were readily distinguishable (Supplementary Fig. 6c). We therefore also consider this dataset for benchmarking.

**Cross-validation experiments and gene reproducibility**. To illustrate the performance of S–E model in real datasets (Supplementary Table 1), we performed cross-validation experiments using the procedure as implemented in scmap: (i) we randomly selected 70% of the cells as the reference set, (ii) we then identified informative genes (based on the reference set) with different feature selection methods respectively, (iii) we further trained the RF classifier[50] using the reference set with only informative genes selected by different methods (cell labels were defined with unsupervised clustering by the original authors), (iv) the remaining 30% cells were considered as query set, and corresponding cell types were predicted with the trained classifier, (v) the classification accuracy was then quantified with the accuracy score[50], which is the similarity between the predicted cell types and the original cell types of the query set, (vi) finally, we repeated this entire procedure for $n = 50$ times for each dataset.

We calculated the reproducibility by intersecting the corresponding sets of variable genes as

$$\text{Reproducibility} = \frac{\text{Geneset}^{m-n}_{\text{replicate}-1} \cap \text{Geneset}^{m-n}_{\text{replicate}-2}}{n},$$

where $m$ denotes the adapted gene selection method and $n$ is the number of top-ranked variable genes.

**Rare cell-type simulation**. We simulated the synthetic scRNA-seq data following the same approach in GiniClust2 (https://github.com/dtsoucas/GiniClust2/blob/master/Rfunction/Generate_Simulated_Data.R), specifying two large 1000 cell clusters, and three rare clusters of 10, 20, and 30 cells, respectively. To test the performance of our method, we applied our S–E model to the raw count data to select informative genes and performed follow-up clustering with standard clustering procedure in Seurat. The R scripts of RaceID3 and GiniClust2 were accessed through https://github.com/dgrun/RaceID3_StemID2_package and https://github.com/dtsoucas/GiniClust2, respectively.

**ROGUE calculation**. By taking advantage of the wide applicability of S–E model to scRNA-seq data, we introduce the statistic ROGUE to measure the purity of a cell population as

$$\text{ROGUE} = 1 - \frac{\sum_{\text{sig}} ds}{\sum_{\text{sig}} ds + K},$$

where the parameter $K$ is used for two purposes: (i) constrain the ROGUE value between 0 and 1, (ii) serve as a reference factor to provide the purity evaluation. Consider a reference dataset with maximum summarization of significant $ds$. We set the value of $K$ to one-half of the maximum. In this way, ROGUE will receive a value of 0.5 when summarized significant $ds$ is equivalent to one-half of the maximum. A cell population with no significant $ds$ for all genes will receive a ROGUE value of 1, while a population with large summarization of significant $ds$ is

supposed to yield a small purity score. We reasoned that Tabula Muris can be considered as such a plausible reference dataset because it comprises cells from 20 organs, which represents a highly heterogeneous population and was sequenced with both 10X Genomics and Smart-seq2 protocols[2]. As the technical variation associated with PCR, which is present in full-length-based but not droplet-based technology, will affect the value of *ds*, we calculated the summarization of significant *ds* of Tabula Muris for both 10X Genomics and Smart-seq2 datasets (Supplementary Fig. 19). Accordingly, we set the default value of *K* to one-half of the summarization, i.e., 45 for droplet-based data and 500 for full-length-based data, receptively. The *K* value can also be determined in a similar way by specifying a different reference dataset in particular scRNA-seq data analyses. Users should be careful when using the default *K* value on datasets of different species, and we recommend the user to determine the *K* value by specifying a highly heterogeneous dataset of that species with the DetermineK function in ROGUE package.

**Silhouette coefficient**. To assess the differences of simulated replicates and the separation of different cell clusters, we calculated the silhouette width[7], which is the ratio of within-cluster to intercluster dissimilarity. Let $a(i)$ denote the average dissimilarity of cell $i$ to all other cells of its cluster A, and let $b(i)$ denote the average dissimilarity of cell $i$ to all data points assigned to the neighboring cluster, whose dissimilarity with cluster A is minimal. The silhouette width for a given cell $i$ is defined as

$$s(i) = \frac{b(i) - a(i)}{\max(a(i), b(i))}.$$

A high $s(i)$ value suggests that the cell $i$ is well assigned to its own cluster but poorly assigned to neighboring clusters.

**Sequencing depth simulation**. Sequencing depth can vary significantly across cells and thus contributes to a substantial technical confounder in scRNA-seq data analysis. To illustrate that ROGUE is robust to sequencing depth, we generated simulated populations, each consisting of two replicates with only differences in sequencing depth (Fig. 4d and Supplementary Fig. 7a). In each simulation, we varied the sequencing depth of the two replicates as

$$\mu_{\text{replicate}-2,i} = \mu_{\text{replicate}-1,i} \cdot \delta, \, i \in \{1, \dots, n\},$$

where $n$ is the number of genes, $\mu$ is the mean expression level, and $\delta \in \{2, 5, 10, 20, 50, 70, 100\}$.

**Generation of simulated cell types**. To demonstrate the potential for ROGUE to guide single-cell clustering, we used NB model as aforementioned to simulate different scRNA-seq datasets, each consisting of three cell types A, B, and C (1000 cells × 10,000 genes each), where A and B were similar subtypes. For the three scenarios shown in Fig. 3a and Supplementary Fig. 15a, d, we introduced 500, 1000, and 800 varied genes between cell-type A and cell-type B/C, respectively, with fold changes drawn from the log-normal distribution ($\mu = 0$ and $\sigma = 2$). In addition, we simulated 100, 100, and 120 highly variable genes between cell-type B and C respectively, with fold changes sampled from a log-normal distribution with $\mu = 0$ and $\sigma = 1$. The results were visualized using *t*-distributed stochastic neighbor embedding (t-SNE) implemented in R package Rtsne.

**Analysis of the fibroblast and B-cell datasets**. To demonstrate the application of ROGUE-guided analysis in identifying pure subpopulations and detecting precise biological signals, we performed re-clustering analysis of the fibroblast and B-cell datasets[31,35]. We filtered out low-quality cells with either <600 expressed genes, over 25,000 or below 600 UMIs. After filtration, a total of 4291 B cells and 1465 fibroblasts were remained. We further applied our S–E model to the raw count data to select informative genes. Although other pioneering methods could be used to calculate size factors[39,40,45], we normalized the gene expression matrices with regularized NB regression in Seurat[23]. The top 3000 genes with maximal *ds* were used for PCA analysis. To remove batch effects between donors, we performed batch correction using BBKNN[51] with the first 50 PCs. Using the leiden clustering approach implemented in scanpy[52], each cell cluster was identified by its principle components. This yielded 11 fibroblast subtypes and 7 B-cell subtypes as shown in Figs. 3d and 4b, which were visualized in 2D projection of UMAP[53] with default parameters. Accordingly, the purity score of each cluster was calculated with the rogue function in our R package. The calculation of ROGUE is based on raw count data and is independent of methods used for normalization, dimensionality reduction, and clustering.

**Pathway and TCGA data analysis**. To characterize and detect the pathway signals in specific fibroblast subtypes, we performed pathway analyses using hallmark pathways from the molecular signature database[54] with GSVA[55]. The TCGA LUAD and LIHC data were used to investigate the prognostic effect of 13 signature genes (Supplementary Table 5) derived from B_C2_ACTB. To eliminate the effects of different B-cell proportions, we normalized the mean abundance level of these 13 marker genes by the expression of MS4A1 gene, and performed subsequent statistical analyses using GEPIA2[56] with default parameters.

**Reporting summary**. Further information on research design is available in the Nature Research Reporting Summary linked to this article.

## Data availability
All datasets used in this study were obtained from their public accessions, and detailed information including the publication citations and accession codes can be found in Supplementary Tables 1–3 and Supplementary References.

## Code availability
Software implementing our approach is available as an open-source R package ROGUE, which can be downloaded at https://github.com/PaulingLiu/ROGUE. All scripts used for benchmarks and figure generation are available at https://github.com/PaulingLiu/ROGUE/tree/master/scripts.

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

## Acknowledgements
We thank Y. He for valuable discussions on this study. This project was supported by Beijing Advanced Innovation Centre for Genomics at Peking University, National Natural Science Foundation of China (31530036, 91742203, and 91942307).

## Author contributions
Z.Z. conceived this study. C.L. and B.L. designed the S–E model. B.L. introduced the algorithm of ROGUE, performed the benchmark testing, analyzed the data, and developed the R package. Z.L., D.W., and X.R. assisted with method development. B.L., C.L., and Z.Z. wrote the manuscript with all the authors' inputs.

## Competing interests
C.L. and Z.Z. are either intern or founder of Analytical Biosciences Limited. The remaining authors declare no competing interests.
