## [Peer Review File · Nature Communications]

Reviewers' Comments:

Reviewer #1:

Remarks to the Author:

In this manuscript Liu et al. present ROUGE an entropy-based metric to assess the purity of cell subpopulations in the context of clustering and the S-E model to select informative genes that can be used to perform clustering.

I think that the ROUGE metric and the S-E model are interesting ideas. However, I am not fully convinced of the superiority of the S-E model based on the proposed analyses. The manuscript lacks several important details and the source code for the proposed comparison is not available. I believe before publication it is necessary to consider the following points:

Major points:

1) In recent single-cell literature several feature selection and clustering strategies have been proposed that are not presented and formally compared with in this paper. Instead, the proposed S-E model is compared only with HVG, Gini and M3DROP. I think this makes this manuscript already outdated. I think it is necessary to compare the proposed S-E model in the context of clustering specifically and formally at least with the following methods:

- The variable gene selection strategy based on the SCTransform procedure (Stuart, Butler, et al., Cell 2019) and their standard clustering procedure proposed.
- GiniClust2, a procedure that integrates the Fano factor with the Gini score (Tsoucas et al. Genome Biology 2018)
- The SC3 clustering workflow (Kiselev et al Nature Methods 2017)
- RaceID3 (Herman et al Nature Methods 2018)

2) What is more worrying is that some of the other methods have been used before the S-E metric, making the entire comparison exercise flawed. For example, in the section: "Processing and analysis of the fibroblast and B cell datasets" I read: "We further normalized the gene expression matrices using regularized negative binomial regression⁴⁴ with the SCTransform function in Seurat²⁵. The top 3000 genes with maximal Pearson residual were used for PCA analysis.". It seems the author used the Seurat strategy to normalize and select informative genes. This raises several questions. Is S-E better than the proposed workflow by Seurat? Do we need to run Seurat and prefilter the genes before applying the S-E selection?

3) Based on the current manuscript it is hard to understand exactly how the analyses on real and simulated data were performed. The source code for the simulations and for the processing of the real data must be provided.

4) Some clustering methods were optimized to recover rare and pure subpopulations. For example, RaceID2/3 and GiniClust 1/2. I think it is necessary to test with the real and simulated datasets (as proposed in these manuscripts) if the genes selected by the S-E model can lead to the recovery of these rare subpopulations. In other words, is S-E effective in uncovering rare subpopulation as demonstrated by the other tools? What is the limit of detection (# cells)? (The smallest proportion presented in the manuscript I believe is 20%)

5) It is not clear what is the ground truth for the real datasets proposed and how the cross-validation is performed. How did you obtain the labels? Are the labels based on marker genes/ FACS sorted populations? Time points? Or based on other computational procedures that have been published?

6) There are several sentences that are too broad and not easy to understand:

- "in "swamping" of useful information causing the deterioration of the results of such variance-based approaches." Please briefly explain.

- "Gini coefficient¹³ could be used to quantify the variation in gene expression, but the limited performance restricts its scalability." It is not clear what is the meaning of scalability here. There are several implementations of the Gini Index that scales to thousands of genes.

- "Although highly informative genes can also be determined by inspecting their weights during multiple iterations of dimensionality reduction¹⁵, such ad hoc approaches are computationally intensive"

Please quantify the computationally intensive. " do not provide independent metrics for gene expression variability." Please explain this better.

7) ROGUE CALCULATION: "and hence provides a universal metric for assessing the purity of given cell clusters". I am not sure setting K based on the Tabula Muris dataset is enough to claim that this metric is universal and generalizable. Is K depending on the model organism used?

Minor Points:

1) It is not clear if the Gini index used was from the original GiniClust or reimplemented by the author.

2) "Using 14 previously published datasets derived from both droplet-based and full-length protocols". It seems that 6 of the 14 datasets are from the same Tabula Muris release.

3) "This ERCC dataset achieved 197 ROGUE ~ 1 with only one significant gene." Please expand on this point.

Reviewer #2:

Remarks to the Author:

An important step in scRNA-seq analysis is to cluster cells into different types, but a practical challenge is to determine the number of clusters in such analysis. Published studies often use ad hoc approaches, and there is no census in terms of when to stop clustering. It is desirable to have a rigorous procedure to guide clustering analysis. The ROGUE index proposed by Liu et al. aimed to tackle this important problem. Through extensive simulations, the authors evaluated the impact of technical variations such as sequencing depth on the performance of ROGUE. The authors also analyzed multiple challenging cancer scRNA-seq data sets, and showed that ROGUE is effective in identifying heterogeneous cell populations, and can help guide clustering analysis. The authors provided a R package, which will facilitate other researchers to utilize their procedure. Below, I list a few suggestions that hope to help the authors to further improve the manuscript.

1) It would be helpful if the authors could provide a methods overview in the beginning of the Results section. This will help readers better understand and interpret the results.

2) Methods section: please define entropy and differential entropy.

3) Methods section: some of the notation is confusing. For example, it doesn't seem necessary to use subscript to represent mean library size.

4) I also don't understand how you got the 2nd last equation on page 10, and what the subscript "i" represent.

5) The ROGUE index depends on the value of "K". Can you provide an intuitive explanation of "K"? Also, can you explain why using different values of "K" for droplet-based and full-length based data sets? How did you pick the value of 45 and 500? It is important to show that ROGUE is robust to the choice of K.

6) How sensitive is ROGUE to the choice of informative genes and the number of cells in a cluster?

7) Will batch effect lead to reduced purity within a cluster? It would be good to evaluate the impact of batch effect on the performance of ROGUE.

8) In addition to cancer data, it would also be valuable if the authors could analyze non-cancer data such as brain data, which include broadly defined cell classes and more refined cell subtypes.

Reviewers' comments:

Reviewer #1 (Remarks to the Author):

In this manuscript Liu et al. present ROUGE an entropy-based metric to assess the purity of cell subpopulations in the context of clustering and the S-E model to select informative genes that can be used to perform clustering.

I think that the ROUGE metric and the S-E model are interesting ideas. However, I am not fully convinced of the superiority of the S-E model based on the proposed analyses. The manuscript lacks several important details and the source code for the proposed comparison is not available. I believe before publication it is necessary to consider the following points:

We thank the reviewer for the enthusiastic comment on our ideas and also greatly appreciate all the constructive suggestions and criticisms, which have helped us further improve the novelty and quality of our manuscript. Regarding the major concern of lacking comprehensive benchmarks, and clarity in description of particular details, here we point out the major improvements in our revised manuscript, especially after integrating all specific comments:

- i> We have released the codes for benchmarking and data analysis on GitHub, (<https://github.com/PaulingLiu/ROGUE/scripts>), and provided clear description of certain details (e.g., how the analyses on real and simulated data were performed), which were not well presented in the previous manuscript.
- ii> We have performed comprehensive comparison to demonstrate the potential superiority of our method by benchmarking S-E model against more competing methods in different contexts, and added new figures for intuitive comparison and visualization.

Major points:

1) In recent single-cell literature several feature selection and clustering strategies have been proposed that are not presented and formally compared with in this paper. Instead, the proposed S-E model is compared only with HVG, Gini and M3DROP. I think this makes this manuscript already outdated. I think it is necessary to compare the proposed S-E model in the context of clustering specifically and formally at least with the following methods:

- The variable gene selection strategy based on the SCTransform procedure (Stuart, Butler, et al., Cell 2019) and their standard clustering procedure proposed.
- GiniClust2, a procedure that integrates the Fano factor with the Gini score (Tsoucas et al. Genome Biology 2018)
- The SC3 clustering workflow (Kiselev et al Nature Methods 2017)
- RaceID3 (Herman et al Nature Methods 2018)

We thank the reviewer for this helpful advice. In our revised manuscript, we have now compared S-E model with HVG, M3DROP, SCTransform, Gini, Fano factor, and the feature selection method implemented in RaceID3 (referred to as RaceID hereafter) in different contexts. Consistent with our previous conclusion, S-E model still shows the best performance in

simulation experiments, reproducibility and cross-validation tasks (Response Figure 1a, Figure 1 and Supplementary Figures 1-4). In addition, we also evaluated the performance of S-E model in the context of unsupervised clustering as suggested by the reviewer. As GiniClust2 utilizes two different clustering approaches (Gini-based DBSCAN and Fano factor-based k-means), it may be not suitable to evaluate the performance of individual feature selection methods. We therefore only applied RaceID3, SC3 and Seurat in the follow-up evaluation. Here we considered five publicly available scRNA-seq datasets (Methods), which include cells from different lines, FACS-purified populations, or well-characterized types, and thus can be considered gold standards. Detailed description of these five datasets are now included in the Methods section. We used adjusted Rand index (ARI) to evaluate these feature selection methods and found that S-E model provides the best performance in terms of ARI in these scenarios (Response Figure 1b, Figure 1h and Supplementary Figure 7).

We believe these analyses substantially improves our manuscript, and thank the reviewer again for the suggestion.

Response Figure 1: The performance of S-E model. (a) Accuracy in identifying differentially expressed genes on data simulated from both NB (left) and ZINB (right) distribution, with subpopulation containing 50% of the cells. The center line indicates the median AUC. The lower and upper hinges represent the 25th and 75th percentiles respectively, and whiskers denote 1.5 times the interquartile range. **(b)** ARI for the dataset comprising 5 cell lines¹ when different feature selection methods were used.

2) What is more worrying is that some of the other methods have been used before the S-E metric, making the entire comparison exercise flawed. For example, in the section: “Processing

and analysis of the fibroblast and B cell datasets” I read: “We further normalized the gene expression matrices using regularized negative binomial regression with the SCTransform function in Seurat. The top 3000 genes with maximal Pearson residual were used for PCA analysis.” It seems the author used the Seurat strategy to normalize and select informative genes. This raises several questions. Is S-E better than the proposed workflow by Seurat? Do we need to run Seurat and prefilter the genes before applying the S-E selection?

We apologize for the unclear description which led to confusion, and would like to clarify that the analysis of fibroblast and B cell datasets was intended to demonstrate the importance of ROGUE in assessing the purity of cell populations, rather than to compare different feature selection methods. For example, we found that the 7 fibroblast clusters of *Lambrechts et al.*² received low ROGUE values. Then we performed re-clustering and identified pure clusters guided by ROGUE. However, we used SCTransform to select genes rather than S-E model when performing clustering, which indeed raises these questions as mentioned by the reviewer. To address this, we have now used S-E model to select genes in our revised manuscript. The newly identified clusters are consistent with our previous version (Response Figure 2), and corresponding follow-up analysis (e.g. pathway enrichment), which are now included in several new figures (Figures 3 and 4), still supports our conclusion that ROGUE-guided clustering is able to identify pure subpopulations.

As for the performance of S-E model and SCTransform, we have now conducted comprehensive comparison in the context of simulation, reproducibility, cross-validation and clustering, and found that S-E model is superior to SCTransform (Response Figure 1, Figure 1 and Supplementary Figures 1-5).

In addition, we have now clearly stated in the Methods section that both S-E model and ROGUE calculation are based on the raw count data and are independent of methods used for normalization, dimensionality reduction and clustering. We have also given detailed description of how the analysis was performed in the revised manuscript (See details in the reply to comment #3). We thank the reviewer for this important comment.

Response Figure 2: Consistency of identified cell clusters using different gene selection methods. Confusion matrix heatmap was used to show the consistency between S-E model-based clusters and SCTransform (SCT)-based clusters for **(a)** B cell and **(b)** fibroblast datasets.

3) Based on the current manuscript it is hard to understand exactly how the analyses on real and simulated data were performed. The source code for the simulations and for the processing of the real data must be provided.

We apologize for the unclear description and have now updated the manuscript with more details about how the analyses were performed. Here we present an example of the real data analysis:

“

Processing and analysis of the fibroblast and B cell datasets

To demonstrate the application of ROGUE-guided analysis in identifying pure subpopulations and detecting precise biological signals, we performed re-clustering analysis of the fibroblast and B cell datasets^{31,34}. We filtered out low-quality cells with either less than 600 expressed genes, over 25,000 or below 600 UMIs. After filtration, a total of 4,291 B cells and 1,465 fibroblasts were remained. We further applied our S-E model to the raw count data to select informative genes. Although other pioneering methods could be used to calculate size factors^{38,39,40}, we normalized the gene expression matrices using regularized negative binomial regression⁴⁴ in Seurat²⁵. The top 3000 genes with maximal ds were used for PCA analysis. To remove batch effects between donors, we performed batch correction using BBKNN⁴⁵ with the first 50 PCs. Using the leiden clustering approach implemented in scanpy⁴⁶, each cell cluster was identified by its principle components. This yielded 11 fibroblast subtypes and 7 B cell subtypes as shown in Fig. 3d and Fig. 4b, which were visualized in 2D projection of UMAP with default parameters. Accordingly, the purity score of each cluster was calculated with the ‘rogue’ function in our R package. The calculation of ROGUE is also based on raw count data and is independent of methods used for normalization, dimensionality reduction and clustering.”

In addition, the source codes for generating the benchmarking results have been released at GitHub: <https://github.com/PaulingLiu/ROGUE/tree/master/scripts>. We thank the reviewer for pointing this out.

4) Some clustering methods were optimized to recover rare and pure subpopulations. For example, RaceID2/3 and GiniClust 1/2. I think it is necessary to test with the real and simulated datasets (as proposed in these manuscripts) if the genes selected by the S-E model can lead to the recovery of these rare subpopulations. In other words, is S-E effective in uncovering rare subpopulation as demonstrated by the other tools? What is the limit of detection (# cells)? (The smallest proportion presented in the manuscript I believe is 20%)

This is an excellent point, and we thank the reviewer for suggesting it. To this end, we first simulated a scRNA-seq dataset that contains three rare clusters (of 10, 30 and 20 cells, respectively) and two common clusters (of 1,000 cells each) using the procedure proposed in GiniClust2 (Response Figure 3a; Methods). Then we clustered these cells with GiniClust2, RaceID3 and S-E model-based Seurat (using S-E model to select genes and Seurat to do clustering), respectively. Of note, all these three methods properly recapitulated the five cell clusters (Response Figure 3b). In addition to simulated data, we also evaluate S-E model using real datasets. Since no gold standard is available for such cases, we considered four cell lines (A549, H2228, H838 and HCC827) of Tian et al.¹, and generated three common cell types (A549, H2228 and H838; of 500 cells for each) and one rare cell type (HCC827; of 20 or 10 cells,

respectively) by down-sampling. Similar to the analysis of simulated data, all the three methods effectively identified both common and rare cell clusters when there were 20 cells of the rare cell type (Supplementary Figure 9a-c). For the dataset with the rare cell type accounting for lower frequency (10 cells, 0.6% of total cells here), RaceID3 and GiniClust2 exhibited their superiority in uncovering the rare cell type as opposed to S-E model-based Seurat (Supplementary Figure 9d-f). Thus, although S-E model is effective in uncovering rare subpopulations to a certain extent, methods specifically developed for this purpose, such as GiniClust and RaceID, may be more appropriate. We have now included these analyses in the revised manuscript.

Response Figure 3: Rare cell type detection. (a) Details of the simulated data with 5 distinct clusters. **(b)** Left, three-dimensional tSNE plot of the GiniClust2 clustering results. Right, Sankey diagram showing the similarity between the true clusters (left-axis) and GiniClust2 clustering results (right-axis). The height of the stratum is proportional to the square root of the number of cells. **(c, d)** Clustering results and performance of RaceID3 **(c)** and S-E model-based Seurat **(d)**, which are analogous to **(b)**.

5) It is not clear what is the ground truth for the real datasets proposed and how the cross-validation is performed. How did you obtain the labels? Are the labels based on marker

genes/FACS sorted populations? Time points? Or based on other computational procedures that have been published?

We apologize for the vague description. The cell labels were defined with unsupervised clustering by the original publication. To evaluate the performance of different feature selection methods, we performed cross-validation experiments using the procedure as implemented in scmap³: (i) we randomly selected 70% of the cells as the reference set, (ii) we then identified informative genes (based on the reference set) with different feature selection methods respectively, (iii) we further trained the random forest classifier using the reference set with only informative genes selected by different methods, (iv) the remaining 30% cells were considered as query set, and corresponding cell types were predicted with the trained classifier, (v) the classification accuracy was then quantified with the *accuracy score*⁴, which is the similarity between the predicted cell types and the original cell types of the query set, (vi) finally, we repeated this entire procedure for n=50 times for each dataset. We have now updated these details in both main text and Methods section.

Although the cell labels of these 14 used datasets were assigned using computational methods and could only be considered 'silver standard', they were collected from different tissues and generated by different sequencing platforms, and thus are expected to be good representative of current scRNA-seq data. In addition, we have also included datasets that can be considered as gold standards in cross-validation tasks (See details in the reply to comment #9), which are able to provide a more convincing evaluation. We thank the reviewer for this helpful comment.

6) There are several sentences that are too broad and not easy to understand:

- "in "swamping" of useful information causing the deterioration of the results of such variance-based approaches." Please briefly explain.

- "Gini coefficient¹³ could be used to quantify the variation in gene expression, but the limited performance restricts its scalability." It is not clear what is the meaning of scalability here. There are several implementations of the Gini Index that scales to thousands of genes.

- "Although highly informative genes can also be determined by inspecting their weights during multiple iterations of dimensionality reduction, such ad hoc approaches are computationally intensive"

Please quantify the computationally intensive. "do not provide independent metrics for gene expression variability." Please explain this better.

We thank the reviewer for pointing this out and have now revised our manuscript as following:

- i) *"In particular, scran¹¹ aims to identify variable genes by comparing variance to a local regression trend. However, the over-dispersion, coupled with the high frequency of dropout events, would often result in the selection of many lowly expressed genes¹²". We have also provided corresponding citation for supporting this.*
- ii) *"Alternatively, although Gini coefficient¹³ could be used to quantify the variation in*

gene expression, it is specially designed for rare cell type identification.”

- iii) “Although highly informative genes can also be determined by inspecting their weights during multiple iterations of dimensionality reduction¹⁵, such ad hoc approaches are computationally intensive, requiring potentially orders of magnitude more time than methods like HVG and M3Drop”. To illustrate this, here we used the Tabula Muris dataset to evaluate the consuming time for HVG, M3Drop and PCA method, which utilizes dimensionality reduction to select genes. For the number of cells, we used 100, 500, 1,000 and 2,000. The time consumption and cell number relationship confirmed the trend that PCA method is not as efficient as HVG and M3Drop regarding the computational process (Response Figure 4). In addition, to avoid any confusion, we have now removed the sentence “do not provide independent metrics for gene expression variability.”

Response Figure 4: Single CPU consuming times for gene selection process with PCA, HVG and M3Drop. Solid lines are loess regression fitting (span = 1.5), implemented with R function `geom_smooth`. The HVG and M3Drop were implemented with the ‘`BrenneckeGetVariableGenes`’ and `M3DropFeatureSelection`’ function in the M3Drop package, respectively. The PCA method was implemented with the script from https://github.com/tallulandrews/M3Drop/blob/master/R/Other_FS_functions.R.

7) ROGUE CALCULATION: “and hence provides a universal metric for assessing the purity of given cell clusters”. I am not sure setting K based on the Tabula Muris dataset is enough to claim that this metric is universal and generalizable. Is K depending on the model organism used?

We agree with the reviewer that setting K based Tabula Muris is not sufficient to claim that the ROGUE metric is universal, and have removed such statements in our revised manuscript.

The parameter K is used for two purposes: (i) constrain the ROGUE value between 0 and 1, (ii) serve as a reference factor to provide the purity evaluation. We set the value of K based on Tabula Muris because it comprises cells from 20 organs and represents a highly heterogeneous population. The K value can also be determined in a similar fashion by specifying a different heterogeneous reference dataset, which is not limited to model organisms. We have now included this point in the revised manuscript and also provided more discussion of the choice of K :

“Users should be careful when using the default K value on datasets of different species, and we recommend the user to determine the K value by specifying a highly heterogeneous dataset of that species with the ‘DetermineK’ function in ROGUE package.”

In addition, we have now demonstrated that ROGUE is robust to the choice of K value within a reasonable range in the revised manuscript (Supplementary Figures 13 and 16). We believe these revisions have improved the rigor of our manuscript, and thank the reviewer for the comment.

Minor Points:

8) It is not clear if the Gini index used was from the original GiniClust or reimplemented by the author.

Thanks for reminding us to clarify the details about the use of the feature selection method. Due to the installing error, we copied the source code of original GiniClust (GiniClust_Preprocess.R, GiniClust_Filtering.R and GiniClust_Fitting.R) (<https://github.com/lanjiangboston/GiniClust/tree/master/Rfunction>), and defined the ‘Gini_fun’ function in our scripts to select genes. We would like to clarify that we didn’t modify the source code and we have checked that every step in our newly defined function is exactly the same as the original GiniClust. We have now updated the details in the Methods section and have also given clear description of the use of the other feature selection methods.

9) “Using 14 previously published datasets derived from both droplet-based and full-length protocols”. It seems that 6 of the 14 datasets are from the same Tabula Muris release.

We thank the reviewer for pointing this out. We chose the 6 datasets from Tabula Muris⁵ by two considerations: (i) they were collected from different tissues, (ii) the annotations of cell clusters are in the controlled vocabulary of the cell ontology⁶ and therefore at a high-confidence level. But the comment by the reviewer prompted us to think that using those 6 datasets might leave readers the impression that our evaluation of different feature selection methods is somehow highly dependent on Tabula Muris. To avoid any confusion, we replaced three of the six datasets with another three datasets comprising different cell lines or well-characterized types (Supplementary Table 1). The newly updated evaluation analysis, which are now included in several new figures (Figure 1e,f and Supplementary Figures 3,4), still supports our previous conclusion that ROGUE achieves the highest accuracy of cross-validation tasks.

10) “This ERCC dataset achieved 197 ROGUE ~1 with only one significant gene.” Please expand on this point.

We have now expanded this point in the revised manuscript as following:

“To illustrate the applicability of ROGUE index to real data, we first considered an ERCC (External RNA Controls Consortium) dataset²⁴, which is a highly controlled experiment dedicated to understanding the technical variability. All 1,015 droplets of this dataset received

the same ratio of ERCC synthetic spike-in RNAs, hence no varied RNAs should be detected in principle. We referred to this dataset as an ideal case of pure cell population and found only one RNA with significant ds. Accordingly, this ERCC dataset achieved a ROGUE value of ~1 as expected, thus confirming its purity.”

We thank the reviewer for this helpful suggestion, which helped us better interpret the results.

Reviewer #2 (Remarks to the Author):

An important step in scRNA-seq analysis is to cluster cells into different types, but a practical challenge is to determine the number of clusters in such analysis. Published studies often use ad hoc approaches, and there is no census in terms of when to stop clustering. It is desirable to have a rigorous procedure to guide clustering analysis. The ROGUE index proposed by Liu et al. aimed to tackle this important problem. Through extensive simulations, the authors evaluated the impact of technical variations such as sequencing depth on the performance of ROGUE. The authors also analyzed multiple challenging cancer scRNA-seq data sets, and showed that ROGUE is effective in identifying heterogeneous cell populations, and can help guide clustering analysis. The authors provided a R package, which will facilitate other researchers to utilize their procedure.

We thank the reviewer for the enthusiastic comments regarding the importance and novelty of our method.

Below, I list a few suggestions that hope to help the authors to further improve the manuscript.

1) It would be helpful if the authors could provide a methods overview in the beginning of the Results section. This will help readers better understand and interpret the results.

We appreciate this helpful advice and have now modified our manuscript by providing an overview in the Results section:

“Overview of ROGUE

As scRNA-seq data can be approximated by negative binomial (NB) or zero-inflated NB (ZINB) distribution^{16,17}, we considered the use of the statistic, S (expression entropy — differential entropy of expression distribution, as defined in Methods), to capture the degree of disorder or randomness of gene expression. Notably, we observed a strong relationship between S and the mean expression level (E) of genes, thus forming the basis for our expression entropy model (S-E model, Fig. 1b-c). Moreover, S is linearly related to E for the Tabula Muris dataset² as expected (Fig. 1b), which is characteristic of current droplet experiments, hence demonstrating the NB nature of UMI-based datasets (Methods). For a heterogeneous cell population, certain genes would exhibit expression deviation in fractions of cells, leading to constrained randomness of its expression distribution and hence the reduction of S . Accordingly, informative genes can be obtained in an unsupervised fashion by selecting genes with maximal S -reduction (ds) against the null expectation (Methods).

To provide a direct purity assessment of putative cell clusters or fluorescence-activated cell sorting (FACS)-sorted populations, here we take advantage of the wide applicability of S-E model to scRNA-seq data and introduce the quantitative measure, ROGUE (Ratio of Global Unshifted Entropy, Fig. 1b and Methods). Intuitively, a cell population with no significant ds for all genes will receive a ROGUE value of 1, indicating it is a completely pure subtype or state. In contrast, a population with maximum summarization of significant ds will yield a purity score of ~ 0 .”

2) Methods section: please define entropy and differential entropy.

We thank the reviewer for pointing this out and have now defined the differential entropy (as we only used differential entropy in our manuscript) as following:

“To capture the degree of disorder or randomness of gene expression, here we considered the use of differential entropy defined as⁴²:

$$H(X) = - \int_{-\infty}^{+\infty} p(x) \cdot \log_2 p(x) dx$$

where X is a continuous random variable and $p(x)$ is the probability density function. Differential entropy is an extension of Shannon entropy, which is used to measure the average surprisal of a continuous probability distribution, and has shown notable performance in our supervised gene selection method E-test⁴³.”

In addition, we have now modified the definition of the reduction of differential entropy to avoid potential confusion as following:

$$ds_i = \text{entropy under } H_0 - \text{average actual entropy} = \ln \bar{X}_i - \frac{\sum_{j=1}^n (\ln X_{ij})}{n} \quad (\text{previous version})$$

$$\downarrow$$

$$ds_i = \text{differential entropy under } H_0 - \text{average actual differential entropy}$$

$$= \mathbb{E}(X_i) - \frac{\sum_{j=1}^n (\ln X_{ij})}{n}$$

As defined in the revised manuscript, $\mathbb{E}(X_i)$ is the expectation of the observed expression of gene i across cells.

3) Methods section: some of the notation is confusing. For example, it doesn't seem necessary to use subscript to represent mean library size.

We thank the reviewer for this helpful comment and have corrected the use of particular notation (also see reply to comment #4). For example, we have now modified the definition of library size as following:

“we considered the library size normalization of each cell defined as the total UMI counts divided by the mean total UMI counts across cells⁴⁰. Accordingly, the expectation of library size factor across cells is equal to 1.”

4) I also don't understand how you got $\bar{X}_i = \frac{\sum_{j=1}^n (X'_{ij} \times s_j)}{n} = \frac{\sum_{j=1}^n (X'_{ij})}{n} \times \frac{\sum_{j=1}^n (s_j)}{n} = \bar{X}'_i$, and what the subscript "i" represent.

We agree with the reviewer that this equation was confusing. To prove that the expectation of the observed expression of gene i across cells (i.e., $\mathbb{E}(X_i)$) is equal to the expectation of the normalized expression, $\mathbb{E}(X'_i)$, we have now modified that equation as following:

“Given the equation (2) and that the gene expression and library size are two independent random variables⁴¹, we have, for gene i ,

$$\mathbb{E}(X_i) = \mathbb{E}(X'_i \times s) = \mathbb{E}(X'_i) \times \mathbb{E}(s) = \mathbb{E}(X'_i) \times 1 = \mathbb{E}(X'_i)$$

where X_i is the vector of the observed expression assigned to gene i and s is the vector of library size assigned to cells.”

The equation (2) is defined as (s_j is the library size of cell j):

$$X'_{ij} = \frac{X_{ij}}{s_j}$$

based on which we can derive that $X_i = X'_i \times s$ and further $\mathbb{E}(X_i) = \mathbb{E}(X'_i \times s)$.

The subscript “i” in the original manuscript is actually “i”, which turns into “i” when there is a top-line “—” above the variable. To avoid any confusion, we have now replaced such notation with more intuitive form. For example, we used $\mathbb{E}(X_i)$ instead of $\overline{X_i}$ in the revised manuscript. We thank the reviewer for this helpful comment, which helped us improve the rigor of our method.

5) The ROGUE index depends on the value of “K”. Can you provide an intuitive explanation of “K”? Also, can you explain why using different values of “K” for droplet-based and full-length based data sets? How did you pick the value of 45 and 500?

We introduce the statistic ROGUE to measure the purity of single cell population as:

$$ROGUE = 1 - \frac{\sum_{sig} ds}{\sum_{sig} ds + K}$$

where the parameter K is used for two purposes: (i) constrain the ROGUE value between 0 and 1, (ii) serve as a reference factor to provide the purity estimation. Consider a reference dataset with maximum summarization of significant ds . We set the value of K to one-half of the maximum. In this way, ROGUE will receive a value of 0.5 when summarized significant ds is equivalent to one-half of the maximum. A cell population with no significant ds for all genes will receive a ROGUE value of 1, while a population with large summarization of significant ds is supposed to yield a small purity score. We reasoned that Tabula Muris can be considered as such a plausible reference dataset because it comprises cells from 20 organs, which represents a highly heterogeneous population and was sequenced with both 10X Genomics and Smart-seq2 protocols⁵. As the technical variation associated with PCR, which is present in full-length-based but not droplet-based technology, will affect the value of ds , we calculated the summarization of significant ds of Tabula Muris for both 10X Genomics and Smart-seq2 protocols (Supplementary Figure 19). Accordingly, we set the default value of K to one-half of the summarization, i.e., 45 for the droplet-based dataset and 500 for the full-length-based dataset, respectively. We have now updated these details in the revised manuscript and given more discussion about the choice of K as following:

“Users should be careful when using the default K value on datasets of different species, and we recommend the user to determine the K value by specifying a highly heterogeneous dataset of that species with the ‘DetermineK’ function in ROGUE package.”

It is important to show that ROGUE is robust to the choice of K.

We agree with the reviewer on this point and have now illustrated it in two aspects:

- i) ROGUE index still provides a sensitive and unbiased measure in response to the degree of cell population purity when different K values were used. The newly updated results can be found in Supplementary Figure 13 and corresponding description is now in Line 184 as following:

“It is evident that the ROGUE index decreased monotonically with the heterogeneity of cell populations (Fig. 2a,b and Supplementary Figs. 11 and 12). ROGUE performed well even when cell populations contained few varied genes (<1%) and “infiltrating” cells (<1%), indicating ROGUE index provides a sensitive and unbiased measure in response to the degree of cell population purity. Use of different values of the reference factor K (Fig. 1b and Methods) yielded vary similar results (Supplementary Fig. 13), suggesting that ROGUE is robust to the choice of parameter K within a reasonable range.”

- ii) ROGUE-guided clustering identified pure cell populations. In the revised manuscript, we tested if ROGUE is still effective in the context of clustering when using different K values, and found that the results (Supplementary Figure 16) still support our previous conclusion that ROGUE could enhance single cell clustering. The corresponding description is as following (Line 246):

“Proper clustering of this dataset should result in three subpopulations, one for each cell type. However, silhouette received the maximum value when cell type A co-clustered with B (Fig. 3b), i.e., when only two clusters were identified, suggesting that such measure is poorly interpretable for cluster purity as opposed to ROGUE, which reached saturation when there were three clusters (Fig. 3c). Repeating the simulation with varied differences in cell type A, B and C yielded equivalent performance for these two methods (Supplementary Fig. 15a-f). Such performance was also observed when different values of the reference factor K were used (Supplementary Fig. 16).”

Thanks for this helpful suggestion.

6) How sensitive is ROGUE to the choice of informative genes and the number of cells in a cluster?

Thanks for reminding us to demonstrate this point. To test how ROGUE varies when different numbers of informative genes are used, here we considered two scRNA-seq datasets: a T cell dataset sequenced with Smart-seq2⁷ and a droplet-based dataset⁵ (Tabula Muris). For the number of features, we used top 50, 100, 200, 500, 1,000 and 2,000 genes (ranked based on *ds*) and found that the heterogeneity score (1-ROGUE) reached saturation when N genes with significant *ds* were selected (Response Figure 5). Use of different values of K yielded the same

results, thus indicating that it is optimal to calculate ROGUE with significant ds . We have now included such analysis in the revised manuscript.

We would like to clarify that we have previously shown that ROGUE is sensitive to both the number of significantly varied genes and “infiltrating” cells, and that the calculation of S (differential entropy) and ROGUE would not be affected by variation in cell number of a given population or cluster:

- i) To investigate the performance of ROGUE, we simulated 1,860 cell populations (2,000 cells x 20,000 genes each) with each population consisting of two cell types. Accordingly, we varied the proportion of informative genes from 0.1% to 50%, with the cell type-size ratio ranging from 1:100 to 1:1. We found that ROGUE could detect the purity reduction even when there were 0.1% informative genes and 1% “infiltrating” cells (Figure 2a,b and Supplementary Figures 11-13).
- ii) To address the potential concern that the number of cells may represent an intrinsic challenge for S and ROGUE calculation, we performed down sampling analysis to test how S was impacted by cell numbers. By calculating the Pearson correlations of S between the randomly down-sampled datasets and the entire datasets, we found the similarity values of >0.99 , suggesting that S and ROGUE calculation would not be affected by variation in cell number (Figure 2c).

Response Figure 5: The relationship between ROGUE and the number of selected informative genes. (a,b) Heterogeneity score (1-ROGUE) versus the number of selected informative genes for the T cell dataset (a) and Tabula Muris dataset (b). The red point corresponds to the heterogeneity score when N genes with significant ds were used.

7) Will batch effect lead to reduced purity within a cluster? It would be good to evaluate the impact of batch effect on the performance of ROGUE.

We appreciate this valuable suggestion. To address this, we studied a dataset of human PBMCs containing multiple distinct cell types⁸. Cells of this dataset were previously split into two

groups—the interferon-beta (IFN-β) stimulated group and the culture-matched control group, thus these two groups could be considered as two batches. Then we applied ROGUE to assess the purity of each cell type (as defined by the original authors) in individual bathes as well as the aggregated cell population (batch 1 and batch 2), and found that ROGUE detected considerable purity reduction in the aggregated group (Response Figure 6a).

As cells of this dataset were collected from eight unrelated individuals, we also tested how ROGUE behaves in estimating the variability (i.e., batch effect) among patients. Here we only used cells from the control group so that the evaluation would not be influenced by IFN-β perturbation. As expected, the aggregated cell populations of all individuals received significantly lower ROGUE values as opposed to patient-specific populations for each cell type (Response Figure 6b). Thus, ROGUE offers a sensitive way to estimate the impact of batch effect. We have now included this point in the revised manuscript.

Response Figure 6: ROGUE detects the impact of batch effect. (a) ROGUE values were shown for batch 1 (the control group), batch 2 (the stimulation group), and aggregated cell population (batch 1 and batch 2) for each cell type. For fair comparison, we equalized the number of cells in each group by down-sampling. The center line indicates the median ROGUE value. The lower and upper hinges represent the 25th and 75th percentiles respectively, and whiskers denote 1.5 times the interquartile range. * $p < 0.05$, ** $p < 0.005$, Student’s t test. **(b)** ROGUE values for individual-specific cell populations and aggregated populations (all individuals). All cells used here were from the control group. Sub-sampling was performed to equalize the number of cells in each group.

8) In addition to cancer data, it would also be valuable if the authors could analyze non-cancer data such as brain data, which include broadly defined cell classes and more refined cell subtypes.

This is an excellent point. To do this, we considered the brain data of Tabula Muris⁵, whose annotations are in the controlled vocabulary of the cell ontology⁶. This dataset identified seven distinct cell types, of which oligodendrocyte and neuron cell types had low ROGUE values of

<0.8, versus ~0.9-1 for the remaining five cell classes (Response Figure 7a). We therefore applied further clustering guided by ROGUE to oligodendrocyte which is of enough cells (n=3,401), and identified 10 refined cell subtypes, each with its specific marker genes (Response Figure 7b,c). Except cluster 6, we found ROGUE values of ~0.9-1 for all the other nine clusters, suggesting their purity (Response Figure 7d). To investigate potential functions of these subtypes, we compared pathway activities and found considerable phenotypic diversity. For example, cluster 5 showed a strong signal of axon guidance signaling (Response Figure 7e), while the neurotrophin signaling pathway was highly activated in cluster 1 (Response Figure 7f). This example further illustrates how ROGUE plays a key role in uncovering pure subpopulations and is now included in our revised manuscript. We thank the reviewer for this helpful suggestion.

Response Figure 7: The application of ROGUE in brain data. (a) ROGUE values of seven distinct brain cell types as defined by the original publication⁵, with each point representing a sample. **(b)** UMAP plot of the 10 identified clusters of oligodendrocytes (n=3,401), color-coded by their associated clusters. **(c)** Expression heatmap of cell type-specific genes of the 10 oligodendrocyte clusters. **(d)** ROGUE values of oligodendrocyte clusters. Each point represents a sample. **(e,f)** Enriched pathways for cluster 5 **(e)** and cluster 1 **(f)**, respectively.

References

1. Tian, L. *et al.* Benchmarking single cell RNA-sequencing analysis pipelines using mixture control experiments. *Nat. Methods* **16**, 479–487 (2019).
2. Lambrechts, D. *et al.* Phenotype molding of stromal cells in the lung tumor microenvironment. *Nat. Med.* **24**, 1277–1289 (2018).
3. Kiselev, V. Y., Yiu, A. & Hemberg, M. scmap: projection of single-cell RNA-seq data across data sets. *Nat. Methods* **15**, 359 (2018).

4. Fabian, P. & Gaël, V. Scikit-learn: Machine learning in Python. *J. Mach. Learn. Res.* **12**, 2825–2830 (2011).
5. Schaum, N. *et al.* Single-cell transcriptomics of 20 mouse organs creates a Tabula Muris. *Nature* **562**, 367–372 (2018).
6. Bakken, T. *et al.* Cell type discovery and representation in the era of high-content single cell phenotyping. *BMC Bioinformatics* **18**, 559 (2017).
7. Zhang, L. *et al.* Lineage tracking reveals dynamic relationships of T cells in colorectal cancer. *Nature* **564**, 268–272 (2018).
8. Kang, H. M. *et al.* Multiplexed droplet single-cell RNA-sequencing using natural genetic variation. *Nat. Biotechnol.* **36**, 89–94 (2018).

Reviewers' Comments:

Reviewer #1:

Remarks to the Author:

I think the authors did a terrific job in addressing all my comments/concerns. The manuscript is significantly improved and I am looking forward to its publication.

Reviewer #2:

Remarks to the Author:

Thanks for doing a great job addressing my previous concerns. The clarity of the paper is significantly improved and the newly added results are convincing. I don't have any additional comments.

REVIEWERS' COMMENTS:

Reviewer #1 (Remarks to the Author):

I think the authors did a terrific job in addressing all my comments/concerns. The manuscript is significantly improved and I am looking forward to its publication.

We thank Reviewer #1 for helping us improve the novelty and quality of this paper.

Reviewer #2 (Remarks to the Author):

Thanks for doing a great job addressing my previous concerns. The clarity of the paper is significantly improved and the newly added results are convincing. I don't have any additional comments.

We thank Reviewer #2 for helping us improve the novelty and quality of this paper.